# GRP78 regulates CD44v membrane homeostasis and cell spreading in tamoxifen-resistant breast cancer

Chun-Chih Tseng[1,8], Ramunas Stanciauskas[2], Pu Zhang[6,8], Dennis Woo[8,9], Kaijin Wu[5], Kevin Kelly[5,8], Parkash S Gill[7,8], Min Yu[8,9], Fabien Pinaud[2,3,4,8], Amy S Lee[1,8]

GRP78 conducts protein folding and quality control in the ER and shows elevated expression and cell surface translocation in advanced tumors. However, the underlying mechanisms enabling GRP78 to exert novel signaling functions at cell surface are just emerging. CD44 is a transmembrane protein and an important regulator of cancer metastasis, and isoform switch of CD44 through incorporating additional variable exons to the extracellular juxtamembrane region is frequently observed during cancer progression. Using super-resolution dual-color single-particle tracking, we report that GRP78 interacts with CD44v in plasma membrane nanodomains of breast cancer cells. We further show that targeting cell surface GRP78 by the antibodies can effectively reduce cell surface expression of CD44v and cell spreading of tamoxifen-resistant breast cancer cells. Our results uncover new functions of GRP78 as an interacting partner of CD44v and as a regulator of CD44v membrane homeostasis and cell spreading. This study also provides new insights into anti-CD44 therapy in tamoxifen-resistant breast cancer.

## Introduction

GRP78 (78 kD glucose-regulated protein, also referred to as BiP or HSPA5) belongs to the heat shock protein 70 (HSP70) family and is a major ER chaperone protein that facilitates protein folding, quality control, and regulates the unfolded protein response (Ni & Lee, 2007; Luo & Lee, 2013; Lee, 2014; Pobre et al, 2019). Overexpression of GRP78 is associated with cancer cell growth, invasion, and drug resistance (Lee, 2014; Cook & Clarke, 2015; Gifford et al, 2016). Atypical translocation of GRP78 to cell surface was observed in various cancer cells and further elevated under stress conditions (Ni et al, 2011; Zhang et al, 2013; Tsai et al, 2015). Cell surface GRP78 (csGRP78) has emerged as a novel regulator of cell surface

signaling, beyond the traditional protein foldase activity as a chaperone protein in the ER (Ni et al, 2011; Zhang et al, 2013; Tsai et al, 2018). Previous studies have highlighted the importance of csGRP78 in cancer cell–matrix adhesion, motility, invasion, and proliferation; however, the underlying mechanisms are just emerging (Misra et al, 2005b; Kelber et al, 2009; Li et al, 2013). Because GRP78 exists on the cell surface primarily as a peripheral protein (Tsai et al, 2015), the identification of plasma membrane proteins that interact with csGRP78 is critical toward understanding how GRP78 is anchored on the cell surface and exerts its signaling functions.

Recently, it was reported that GRP78 facilitated chemo-radioresistance and invasion in head and neck cancer (HNC) cells exhibiting molecular characteristics (CD24[-]CD44[+]) of HNC stem cells (Chiu et al, 2013). In addition, GRP78 knockdown in HNCs suppressed stem cell regulatory proteins, Oct-4 and Slug, and transformed cell morphology into rounder cell shape (Chiu et al, 2013). CD44 is a type I transmembrane glycoprotein known to facilitate cell adhesion, spreading, migration, invasion, ROS defense, and drug resistance in a variety of cancer types (Misra et al, 2005a; Ishimoto et al, 2011; Zoller, 2011; Montgomery et al, 2012; Hiraga et al, 2013). It is widely used as a cancer stem cell marker in subtypes of cancers including breast (Al-Hajj et al, 2003; Liu et al, 2010; Yan et al, 2015) and serves as the major receptor of hyaluronan (Ghatak et al, 2010). It can also bind to a wide range of ECM components, including metalloproteinases, collagen, laminin, chondroitin sulfate, and fibronectin (Zoller, 2011). CD44 is a highly heterogeneous glycoprotein; it can be regulated by alternative splicing and posttranslational modifications (Zoller, 2011; Yae et al, 2012). CD44 variant isoforms are created by alternative splicing through incorporation of variable exons into the extracellular juxtamembrane region. CD44 standard isoform (CD44s) lacks variable exons. In addition to its role as a cell surface receptor, CD44 variant isoforms can function as a co-receptor that binds FGF2, HGF, VEGF, and osteopontin and present them to their receptors (Zoller, 2011). It has been reported that breast cancer stem-like cells expressing CD44 variant isoforms exhibited enhanced metastatic capacity (Yae et al,

[1]Department of Biochemistry and Molecular Medicine, University of Southern California, Los Angeles, CA, USA    [2]Department of Biological Sciences, University of Southern California, Los Angeles, CA, USA    [3]Department of Chemistry, University of Southern California, Los Angeles, CA, USA    [4]Department of Physics and Astronomy, University of Southern California, Los Angeles, CA, USA    [5]Department of Medicine/Division of Hematology, University of Southern California, Los Angeles, CA, USA    [6]Department of Molecular Microbiology and Immunology, University of Southern California, Los Angeles, CA, USA    [7]Department of Pathology, University of Southern California, Los Angeles, CA, USA    [8]University of Southern California Norris Comprehensive Cancer Center, University of Southern California, Los Angeles, CA, USA    [9]Department of Stem Cell Biology and Regenerative Medicine, University of Southern California, Los Angeles, CA, USA

Correspondence: amylee@usc.edu

2012). CD44 containing variable exons 3 to 10 (CD44v3-10) instead of CD44v8-10 or CD44s is correlated with poor prognosis of breast cancer patients (Hu et al, 2017). Collectively, CD44 is a critical regulator of cytoskeletal dynamics, cell motility, migration, and invasion in normal development and cancer progression (Senbanjo & Chellaiah, 2017).

In searching for GRP78 interactive partners on the cancer cell surface, we recently discovered that CD44v3-10 (hereinafter CD44v) forms complex with GRP78, and they co-localize on the cell surface of tamoxifen-resistant MCF7 cells (MCF7-LR) (Tseng et al, 2019). In this study, we further explored their interaction and functional significance using biochemical assays, time-lapse total internal reflection fluorescence/photoactivated localization microscopy (TIRF/PALM) imaging, and functional studies. We discovered that CD44v can directly bind to GRP78 in vitro, and they exhibit co-diffusion and co-confinement in plasma membrane nanodomains in MCF7-LR cells. We further showed that monoclonal antibodies against csGRP78 can effectively reduce cell surface expression of CD44v and suppress cell spreading in MCF7-LR cells. Our study uncovers a new mechanism for csGRP78 to regulate behaviors of tamoxifen-resistant breast cancer cells at least in part via modulating CD44 protein expression.

# Results

## Co-expression and co-localization of CD44v and GRP78 in breast cancer

It has been reported that tamoxifen-resistant MCF7-LR breast cancer cells exhibited elevated expression of GRP78 (Zhang et al, 2013; Cook & Clarke, 2015) and CD44 (Hiscox et al, 2012; Bellerby et al, 2016). To characterize their interactions in such cells, we first confirmed that MCF7-LR cells expressed CD44v, a single-pass transmembrane protein containing variable exons 3 to 10 (Fig 1A), using RT–PCR and DNA sequencing (Fig 1B and Supplemental Data 1). At the protein level, the expression of CD44v in MCF7-LR cells was confirmed by Western blotting using the monoclonal antibody specifically against CD44 variable exon 3. The observed molecular sizes of CD44v ranging from 100 to 250 kD were consistent with those previously reported (Hu et al, 2017) (Fig 1C). To extend our studies in patient-derived cells, we used circulating tumor cells (CTCs), BRx-68 and BRx-07, derived from breast cancer patients, which are considered as metastatic precursors (Alix-Panabieres et al, 2007; Aceto et al, 2014). The molecular sizes of CD44v detected in CTCs were similar to MCF7-LR cells (Fig 1C). To further confirm the specificity of the anti-CD44v antibody and the identity of CD44v, we overexpressed HA-tagged CD44v (vHA) and detected a major 250-kD protein band similar to the size of endogenous CD44v in Fig 1C (Fig 1D). Furthermore, a positive correlation of csGRP78 and CD44v expression levels was observed in MCF7-LR cells as demonstrated by flow cytometry, with the specificity for the signals confirmed by the absence of primary antibody (M.O.M. control) and isotype control staining (Fig 1E).

Next, we investigated the spatial distribution of csGRP78 and CD44v using immunofluorescent (IF) staining and confocal microscopy. We observed endogenous csGRP78 and CD44v exhibited punctate co-localization in nonpermeabilized migratory MCF7-LR cells, similar to that reported for nonmigratory cells (Tseng et al, 2019) (Fig 1F, left panels). The co-localization of GRP78 and CD44v from Z-stack images covering whole cells from nine independent image areas containing 23.5 cells was quantified by Mander's overlap coefficient (O.C.). The results showed that high percentage of csGRP78 co-localized with CD44v (high M1 value), and partial CD44v co-localized with csGRP78 (Fig 1F, right panel). The nonpermeabilized ex vivo cultured CTCs, BRx-68 and BRx-07, also exhibited punctate staining and co-localization of GRP78 and CD44v at the plasma membrane (Fig 1G, arrows; left panels and Fig S1A). The co-localization of GRP78 and CD44v from Z-stack images covering whole cells from five independent image areas containing 70 (BRx-68) or 307 (BRx-07) cells was quantified by Mander's O.C. in each cell type. The results showed that large percentage of GRP78 co-localized with CD44v in both cell types (intermediate to high M1 values), with lower percentage of GRP78 co-localized with CD44v in BRx-68 cells and partial CD44v co-localized with GRP78 (Fig 1G, right panels). The control stainings for immunocytochemistry ruled out background from first primary antibody and secondary antibodies (Fig S1B and C, respectively). Collectively, these results established that csGRP78 co-localizes with CD44v at the plasma membrane of MCF7-LR cells and patient-derived breast cancer metastatic precursor cells, BRx-68 and BRx-07.

## The CD44v3-10 isoform is predominant in MCF7-LR cells and its extracellular domain can directly bind to GRP78 in cell-free systems

The CD44v isoform, CD44v3-10, identified in this study has been correlated with poor prognosis of breast cancer patients (Hu et al, 2017). To examine whether CD44 containing variable exon 3 is the predominant isoform in MCF7-LR cells, we investigated CD44 protein expression patterns in these cells using two antibodies, one against a common epitope of CD44 (A1351) and the other specifically against the variable exon 3 (BMS144) for co-immunostaining and confocal microscopy (Fig S2). We observed that substantial amount of A1351 signal co-localized with BMS144 signal, indicating most of CD44 isoforms on the cell surface contain variable exon 3. Of note, a population of BMS144 signal (CD44v) did not co-localize with A1351 signal in MCF7-LR cells. This is likely because of differential post-translational modifications, as CD44 is one of the most heavily posttranslationally modified proteins at the plasma membrane.

Next, we investigated if CD44v can directly bind to GRP78. We recently reported that the recombinant GST-tagged full-length GRP78 (FL) and the carboxyl-terminal half of GRP78 (C) can form complex with CD44v in whole-cell lysate (WCL) of 293T cells transfected with CD44v-HA expression plasmid in in vitro GST pull-down system (Tseng et al, 2019). However, whether these two proteins form a complex through direct or indirect binding was not clear. To address this issue, we tested if the recombinant FL protein can directly bind to recombinant CD44 protein in cell-free assay. First, we confirmed the quality of the recombinant FL protein by GST pull-down assay. Briefly, we constructed bacterial expression plasmids for GRP78 FL and mutants containing only the amino-terminal (N) or the carboxyl-terminal half (C) (Fig 2A) and

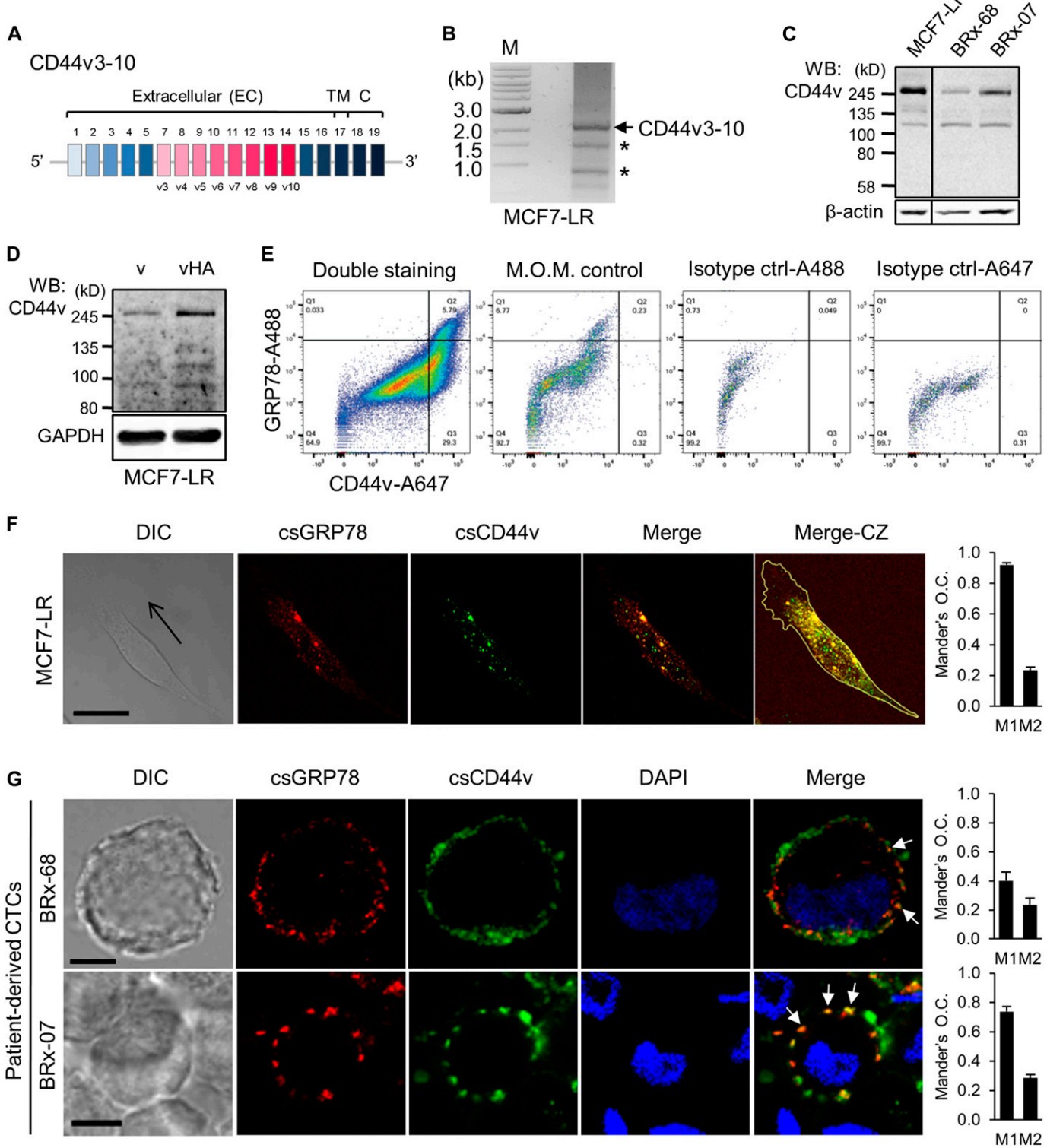

**Figure 1. Co-localization of GRP78 and CD44v in breast cancer cells.**
**(A)** Gene structure of CD44 containing variable exons 3 to 10 (CD44v3-10). Human CD44 contains 19 exons, and CD44v3-10 lacks exon 6. **(B)** Gel electrophoresis of RT–PCR products from MCF7-LR cells. The DNA extracted from the gel bands were cloned into pcDNA3 vector and subjected to DNA sequencing. The arrow indicates the position of CD44v3-10. The asterisks denote nonspecific bands. **(C)** Western blot analysis of WCLs prepared from breast cancer cell lines indicated on top using the antibody specifically targeting CD44v3 exon. β-actin served as a loading control. **(D)** Western blot analysis of WCLs from MCF7-LR cells transfected with HA-tagged CD44v3-10 (vHA) expression plasmid or backbone pcDNA3 vector (v) using the antibody specifically targeting CD44v3 exon. GAPDH served as a loading control. **(E)** Flow cytometry dot plots representing IF double staining results of endogenous levels of cell surface GRP78 and CD44v in MCF7-LR cells. GRP78 and CD44v were detected by MAb159 and anti-CD44v3

mammalian expression plasmid for HA-tagged CD44v (CD44v-HA) (Fig 2B). The recombinant GST-tagged GRP78 proteins were prepared from *Escherichia coli* (BL21) cells, and the purity and identity of these recombinant proteins were confirmed by colloidal blue staining and Western blot analysis, respectively (Fig 2C). CD44v-HA was expressed in 293T cells known to express low endogenous level of CD44 (Ishimoto et al, 2011) and its identity was confirmed by Western blot analysis using anti-CD44 antibody (Fig 2D). Interestingly, the observed 130 kD molecular size of CD44v-HA in 293T cells was different from the breast cancer cells (Fig 1C and D) likely because of cell type–specific posttranslational modifications. We then performed GST pull-down assay and confirmed the quality of recombinant FL for the in vitro assays because we observed that the FL showed similar capacity as C to form complex with CD44v-HA expressed in the WCL of 293T cells (Fig 2E) as previously described (Tseng et al, 2019). We then constructed mammalian expression plasmid for polyhistidine-tagged extracellular region of CD44v (CD44v-EC-His) (Fig 2B) and prepared the recombinant CD44v-EC-His protein from 293T cells. The purity of the recombinant protein was confirmed by colloidal blue staining (Fig 2F). Then, we performed the in vitro direct binding assay, and the binding of FL, but not GST alone, to CD44v-EC-His was observed (Fig 2G). Collectively, these results suggest that the extracellular domain of CD44v can directly bind to GRP78 in cell-free systems.

### Dual-color single-particle tracking reveals the interaction and co-confinement of GRP78 and CD44v in plasma membrane nanodomains

To further characterize the interaction of GRP78 and CD44v with high spatial resolution, we used single-molecule tracking and TIRF microscopy. We constructed two expression plasmids with GRP78 fused to the photoactivatable Tag-red fluorescent protein (PATagRFP-GRP78) and CD44v fused to the photoactivatable green fluorescent protein (CD44v-PAGFP) for dual-color single-particle tracking by PALM (sptPALM) (Subach et al, 2010) (Fig 3A). Proper cell expression of both fusions was confirmed by Western blot analysis (Fig 3B and C). When imaged by TIRF on live MCF7-LR cells, lateral diffusion and partial co-localization of GRP78 and CD44v were observed in the plasma membrane as shown in the conventional TIRF images and the images of super-resolved positions for both proteins (Fig 3D).

The distribution of diffusion coefficients determined by the analysis of individual mean square displacements (MSDs) for CD44v

and GRP78 indicated that both proteins display rather heterogeneous lateral mobility at the cell surface, with GRP78 diffusing significantly slower than CD44v (Fig 3E). A more detailed diffusion analysis by probability distribution of the squared displacement (PDSD) revealed two equal populations of fast (49%) and slow (51%) diffusing CD44v, respectively, undergoing free diffusion at the cell surface ($D_{1-CD44v}$ = 0.213 ± 0.004 $\mu m^2/s$) or confined diffusion ($D_{2-CD44v}$ = 0.016 ± 0.001 $\mu m^2/s$) in membrane nanodomains having a mean radius size of 74 nm (Fig 3F). These diffusion coefficients, the respective fractions of fast and slow diffusing populations, and confinement in plasma membrane nanodomains are in good agreement with previous single-particle tracking and fluorescence recovery after photobleaching of CD44 in other cells (Jacobson et al, 1984; Wang et al, 2014; Freeman et al, 2018). Moreover, the observed lateral diffusion coefficients for CD44 closely resemble our previous report for CD4 (Pinaud & Dahan, 2011), which like CD44 displayed the same ability to associate with glycosphingolipid-rich plasma membrane, consistent with the ability of CD44 to cluster in glycosphingolipid-rich plasma membrane domains (Ilangumaran et al, 1998; Wang et al, 2014).

Using the same PDSD analysis on GRP78, we also identified two confined diffusive behaviors of GRP78. A minority fraction of fast diffusing GRP78 (24%, $D_{1-GRP78}$ = 0.104 ± 0.006 $\mu m^2/s$) was confined in membrane domains 280 nm in radius, whereas a majority fraction (76%) diffused slower ($D_{2-GRP78}$ = 0.00095 ± 0.0006 $\mu m^2/s$) in nanodomains 65 nm radius, a size similar to the confinement domain of slow diffusing CD44v (Fig 3F). From an observation of GRP78 and CD44v diffusion trajectories, we detected cases where both fluorescent protein fusions undergo correlated lateral diffusion and co-confinement in the same nanodomains within the plasma membrane (Fig 3G and Videos 1, 2, and 3), consistent with the interactions and binding between GRP78 and CD44v observed in biochemical assays. The similarity in nanodomain size for both slow diffusing populations of GRP78 and CD44v (65 versus 74 nm) also suggested that the proteins might indeed co-diffuse in shared membrane nanodomains. To further assess how GRP78 influences the dynamics and the confinement of CD44v, endogenous GRP78 was down-regulated by siRNA (si78) and CD44v was again tracked at the plasma membrane of MCF7-LR cells. As shown in Fig 3H and I, reduced expression of GRP78 led to significantly faster diffusion of the free and fast diffusing subpopulation of CD44v (56%, Fig 3H). Interestingly, GRP78 knockdown resulted in significantly slower diffusion for the slow diffusing subpopulation of CD44v (44%, Fig 3H). In MCF7-LR cells treated with a control siRNA (sictrl), no significant differences in

---

antibodies, respectively. Thresholds were set according to control staining. Mouse-on-mouse (M.O.M.) control staining was performed with the same protocol as double staining but lacked the primary antibody against CD44v. The cells in isotype IgG controls were incubated with control IgG and Alexa Fluor 488–conjugated or Alexa Fluor 647–conjugated secondary antibodies. Cell number (n) analyzed in the study: 100,000 for doubled-stained cells; ~10,000 for each control. **(F)** Left panels: Immunofluorescence and confocal images showing the distribution and co-localization of GRP78 (red) and CD44v (green) on the cell surface of nonpermeabilized unipolar MCF7-LR cells. GRP78 and CD44v were detected by MAb159 and anti-CD44v3 antibodies, respectively. Arrow indicates direction of migration. Thickness of single IF image section: 0.8 $\mu m$. Cell periphery was outlined with the white line. Scale bar, 20 $\mu m$. Right: Mander's O.C.: M1 is the contribution of GRP78 to the co-localized area, whereas M2 is the contribution of CD44v. Data represent mean ± SEM. Number of analyzed independent image areas (A) and cells (N): A/N = 9/23.5. **(G)** Left panels: Immunofluorescence and confocal images representing the distribution and co-localization of GRP78 (red) and CD44v (green) on the cell surface of nonpermeabilized breast cancer patient–derived BRx-68 and BRx-07 CTCs. GRP78 and CD44v were detected by MAb159 and anti-CD44v3 antibodies, respectively. Representative co-localizations were indicated by arrows. Thickness of image section: 0.3 $\mu m$. The nuclei were stained by DAPI in blue. DIC: differential interference contrast. Scale bars, 5 $\mu m$. Right panels: Mander's O.C.: M1 is the contribution of GRP78 to the co-localized area, whereas M2 is the contribution of CD44v. Data represent mean ± SEM. Number of analyzed independent image areas (A) and cells (N): A/N = 5/70 (BRx-68); 5/307 (BRx-07). C, cytosolic, CZ, compressed z-stacks; DIC, differential interference contrast; M, marker; TM, transmembrane.

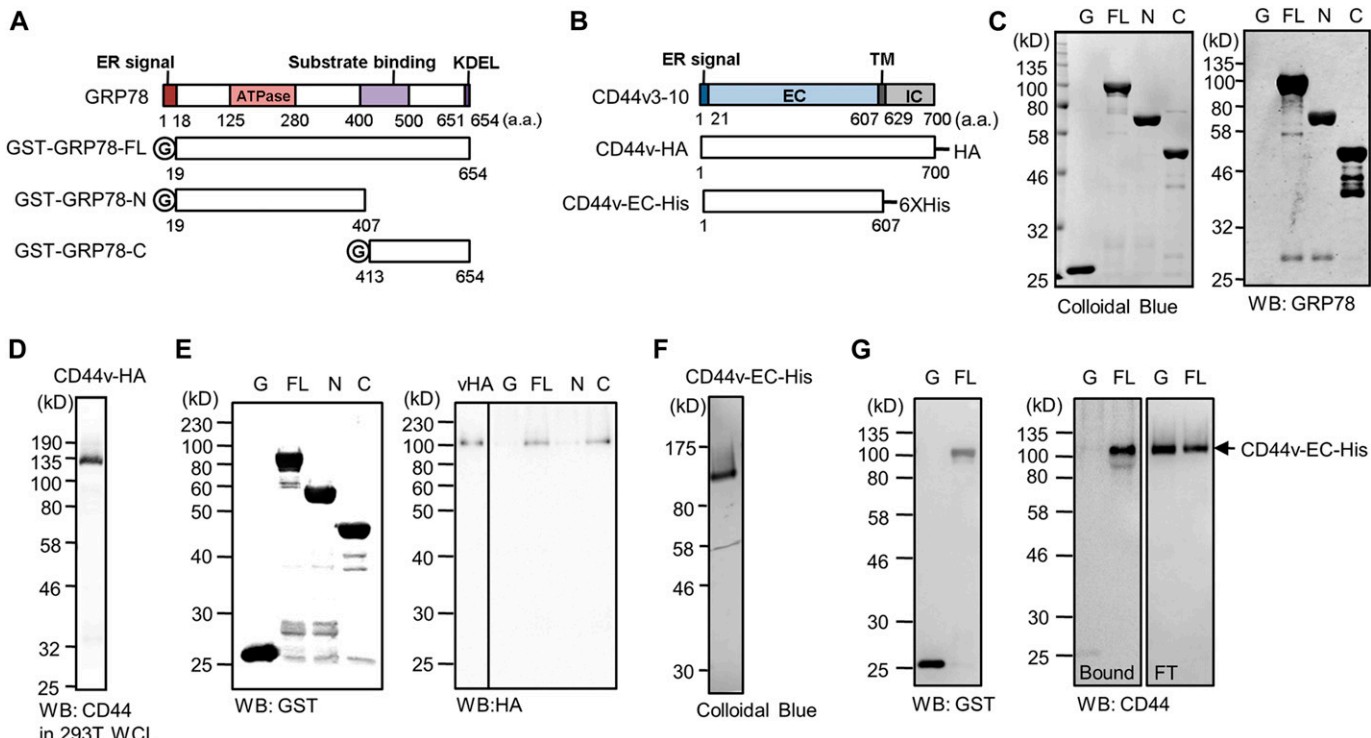

**Figure 2. Direct binding of GRP78 to CD44v in vitro.**
**(A)** Schematic illustration of human GRP78 protein containing an ER signal sequence, ATPase domain, substrate binding domain, and KDEL Golgi-to-ER retrieval motif. The schematic drawings also show GST-tagged FL GRP78 and truncated mutants (N and C). **(B)** Schematic representation of human CD44 protein containing variable exons 3 to 10 (CD44v3-10) and expression constructs for HA-tagged CD44v3-10 (CD44v-HA) and hexahistidine-tagged extracellular region of CD44v3-10 (CD44v-EC-His). **(C)** Left: SDS–PAGE analysis followed by colloidal blue staining of recombinant GST and GST-tagged FL GRP78 and truncated mutants (N and C). 2 μg was loaded for each protein. Right: Western blot analysis of recombinant GST and GST-tagged GRP78 FL, N and C using antibodies against GRP78 (76-E6 and then N-20). **(D)** Western blot analysis of WCL prepared from 293T cells transfected with CD44v-HA expression plasmid using the antibody against CD44 (102111). **(E)** Western blot analysis of samples from in vitro GST pull-down assay. Recombinant GST, GST-tagged GRP78 FL, or N or C purified from *E. coli* (BL21) was incubated with the WCL prepared from 293T cells transfected with CD44v-HA expression plasmid. Input GST and GST-tagged GRP78 proteins are shown in the left panel. Input CD44v-HA is shown in the leftmost lane in the right panel. Three biological repeats showed similar results. **(F)** SDS–PAGE analysis followed by colloidal blue staining for recombinant CD44v-EC-His (2 μg). **(G)** Western blot analysis of samples from in vitro direct binding assay. GST or GST-tagged FL GRP78 proteins purified from *E. coli* (BL21) was incubated with freshly prepared recombinant CD44v-EC-His proteins purified from 293T cells. Input GST and GRP78 proteins are shown in the left panel. The arrow indicates CD44v-EC-His. Flow through (FT) shows unbound fraction. CD44v-EC-His was detected by the anti-CD44 (102,111) antibody. Four biological repeats showed similar results. EC, extracellular; G, GST; IC, intracellular; TM, transmembrane.

CD44v diffusion coefficients were observed compared with non-treated (NT) cells (Fig 3H). Together, these data suggest that GRP78 interacts with CD44 at the plasma membrane nanodomains, and the expression of GRP78 impacts the diffusive behavior and the interaction of CD44 with nanometer size domains.

### GRP78 and CD44 are required for F-actin integrity and cell spreading in MCF7-LR cells

We next used siRNAs against *GRP78* or *CD44* to investigate the functional importance of GRP78 and CD44 in MCF7-LR cells. The efficacy of the four siRNAs (two against each gene) has been previously validated (Godar et al, 2008; Higo et al, 2010; Chen & Lee, 2011; Ishimoto et al, 2011). For the *GRP78* knockdown experiments, si78(1) and si78(2) effectively reduced GRP78 protein levels by about 80% and 70%, respectively, compared with control siRNA, and they did not affect the protein levels of HSP70, a related chaperone protein that shares 62% amino acid sequence identity with GRP78 (Fig 4A). The experimental results were supported by the DNA sequence alignment where it

showed that the siRNA target sequences of *GRP78* lack significant sequence homology to *HSP70* (Fig 4B). The knockdown efficiency of siCD44(1) and siCD44(2) was confirmed by IF staining and epifluorescent imaging using antibodies against a common region of CD44 and specifically against the v3 exon. For both antibodies, we observed about 45% to 60% reduction of CD44 levels compared with control siRNA, with siCD44(1) being more effective (Fig S3A and B).

Down-regulation of either GRP78 or CD44 by siRNAs resulted in rounder cell shape (Fig 4C). As dynamic assembly of filamentous actin (F-actin) network controls cell shape change, we investigated if GRP78 and CD44 can regulate F-actin cytoskeleton. To address this issue, we visualized F-actin network with rhodamine phalloidin staining and epifluorescent imaging (Fig 4D). We observed that knockdown of GRP78 or CD44 by siRNAs suppressed the formation of long F-actin bundles observed in the control (Fig 4D, arrows). To further explore the functional significance, we performed cell adhesion assay and found about 30–65% reduction of cell attachment upon knockdown of GRP78 or CD44 compared with the control (Fig 4E and F). Furthermore, the capability of cell spreading

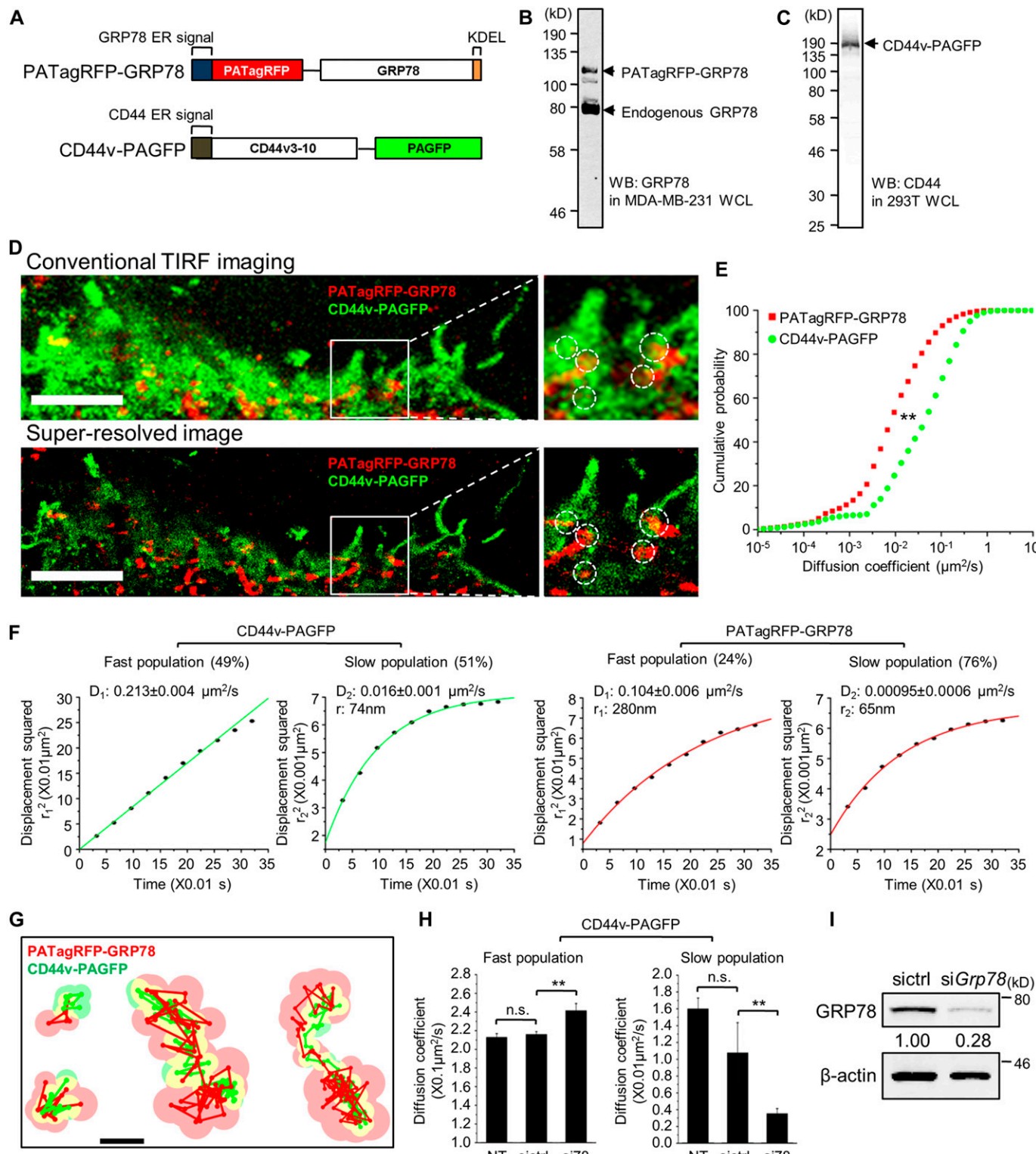

**Figure 3. Dual-color single-particle tracking reveals the interaction and co-confinement of GRP78 and CD44v in plasma membrane nanodomains.**
**(A)** Schematic illustration showing the constructs of PATagRFP-GRP78 and CD44v-PAGFP. **(B)** Western blot analysis of WCL prepared from MDA-MB-231 cells transfected with PATagRFP-GRP78 expression plasmid using antibody against GRP78 (MAb159). **(C)** Western blot analysis of WCL prepared from 293T cells transfected with CD44v-PAGFP expression plasmid using the antibody against CD44 (102111). **(D)** Conventional TIRF (upper panels) and super-resolved (lower panels) images of PATagRFP-GRP78 (red) and CD44v-PAGFP (green) in MCF7-LR cells. Boxed regions are enlarged in the right panels. Dashed circles highlight co-localized area. Scale bars, 5 $\mu$m. **(E)** The distribution of diffusion coefficients was determined by the analysis of individual MSDs for CD44v-PAGFP (n = 51,540 trajectories) and PATagRFP-GRP78 (n = 37,269 trajectories). **P < 0.01 by Kolmogorov–Smirnov test. **(F)** Diffusion analysis by PDSD showing quantifications of square displacement per second (D) and radius of

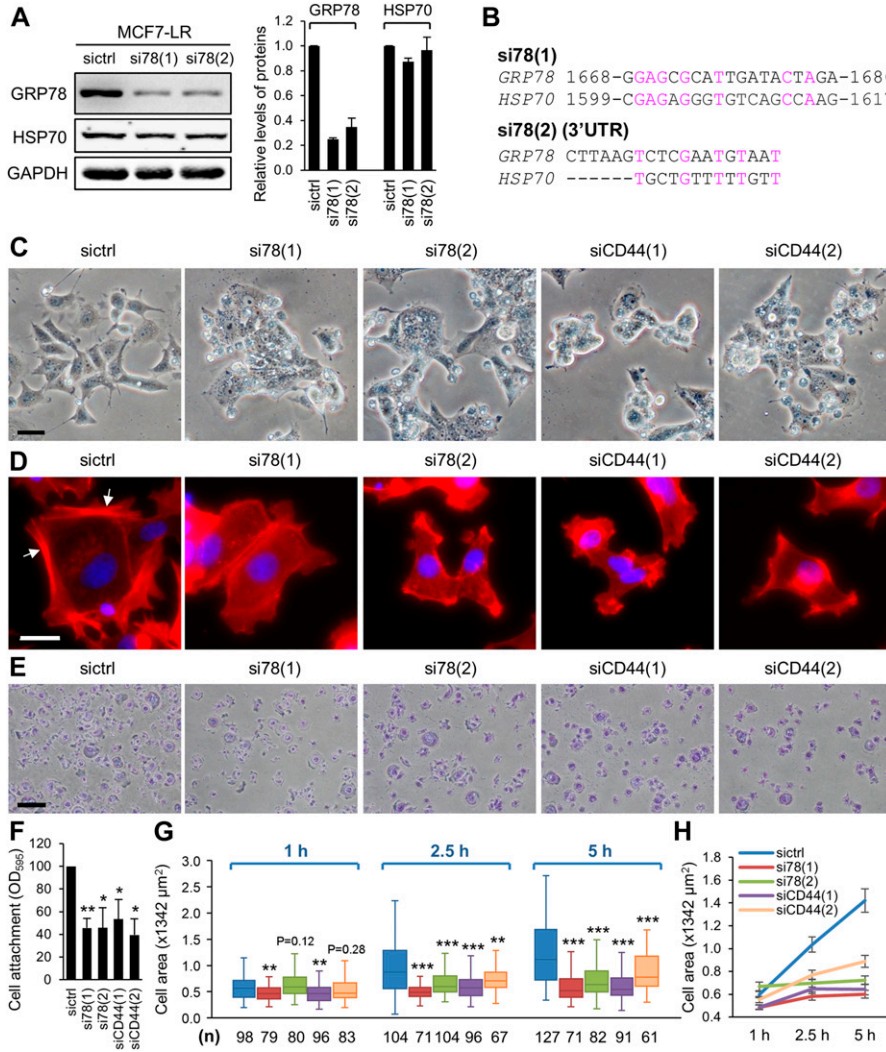

**Figure 4. Knockdown of GRP78 or CD44 alters cell morphology, reduces cell attachment, and impedes cell spreading in MCF7-LR cells.**

**(A)** Left: Western blot analysis of WCLs prepared from MCF7-LR cells transfected with control siRNA (sictrl) or siRNAs targeting *GRP78* coding sequence, si78(1), or 3′ untranslated region (3′-UTR), si78(2), using antibodies against GRP78 (MAb159) or HSP70 (C92F3A-5). GAPDH served as a loading control. Right: Quantification of relative levels of GRP78 and HSP70. Data represents mean ± SEM from three biological repeats. **(B)** Sequence comparison of *GRP78* siRNA target sites on human *GRP78* and *HSP70* genes. **(C)** Bright-field micrographs showing the morphology of MCF7-LR cells transfected with sictrl or siRNA targeting *GRP78* or *CD44*. siCD44(1) targets 3′-UTR and siCD44(2) targets the common coding sequence. Scale bar, 100 μm. **(D)** Epifluorescent micrographs showing the F-actin organization of MCF7-LR cells transfected with sictrl or siRNA targeting *GRP78* or *CD44*. The arrows indicate the F-actin bundles. Red, F-actin labeled with rhodamine phalloidin; blue, nuclei stained by DAPI. Scale bar, 50 μm. **(E)** Cell attachment assay. MCF7-LR cells were transfected with sictrl or siRNA targeting *GRP78* or *CD44* for 60 h before re-seeding onto collagen I–coated culture plates for 1 h. Attached cells were visualized by crystal violet staining. Scale bar, 50 μm. **(F)** Quantification of relative levels of cell attachment described in panel (E). Crystal violet staining of adherent cells was dissolved in 100% methanol, and the OD was measured at 595 nm. Data represent mean ± SEM from three biological repeats. *P < 0.05, **P < 0.01 (*t* test). **(G)** Kinetic measurement of cell spreading area. MCF7-LR cells were transfected with sictrl or siRNA targeting *GRP78* or *CD44* for 60 h before re-seeding onto collagen I–coated culture plates for the indicated times. Cells were stained with rhodamine phalloidin and visualized by epifluorescent microscopy. Cell areas were quantified by the FIJI-Image J software, and the results were represented by the box and whisker plot. n, number of analyzed cells; h, hour. **P < 0.01, ***P < 0.001 (*t* test). **(H)** Data obtained from panel (G) were represented by mean ± SEM for each condition.

was significantly decreased upon knockdown of GRP78 or CD44 compared with the control (Fig 4G and H). These results indicate that GRP78 and CD44 are required for F-actin integrity, cell attachment, and cell spreading consistent with the reported critical roles of CD44 in regulating F-actin network (Bourguignon et al, 2005; Acharya et al, 2008).

## Targeting cell surface GRP78 reduces cell surface expression of CD44v and perturbs F-actin organization and cell spreading

To further investigate if the observed cellular functions of GRP78 and CD44 revealed by siRNA treatment can be at least in part

attributed to csGRP78, we performed antibody screening for commercially available antibodies and then identified a rat monoclonal anti-GRP78 antibody (76-E6) that had the highest potency in altering F-actin structure and cell morphology in 24 h after treatment in MCF7-LR cells (Fig S4A). Importantly, we found that treatment of MCF7-LR cells with 76-E6 led to substantial reduction of CD44v protein levels compared with the IgG and NT controls as evidenced by flow cytometry and Western blot analysis (Fig 5A and B). As membrane-associated matrix metalloproteinase (MMP) has been shown to cleave CD44 (Okamoto et al, 1999), we tested if the reduction of CD44v upon the treatment of 76-E6 was because of proteolytic cleavage by MMPs. To address this issue, we

nanodomain (r) in each fast and slow population of CD44v-PAGFP and PATagRFP-GRP78. **(G)** Examples of CD44v-PAGFP (green) and PATagRFP-GRP78 (red) co-diffusion at the plasma membrane. Circles represent the localization error at each position for individual CD44v-PAGFP (light green) and PATagRFP-GRP78 (light red) along their respective path of diffusion. Yellow areas indicate effective co-localization within position error. Scale bar, 100 nm. **(H)** The diffusion coefficients of CD44v-PAGFP showing fast and slow populations in NT cells and cells treated with control (sictrl) or *Grp78* (si78) siRNA. Trajectory number (n) analyzed in the study: n = 102,957 (sictrl), n = 164,741 (si78), **P < 0.01 (*t* test); ns, not significant. The si78 sequence used is si78(1) described in Table S3. **(I)** Western blot analysis of WCL prepared from MCF7-LR cells transfected with control (sictrl) or *Grp78* (si*Grp78*) siRNA for the diffusion tracking analysis. β-actin served as a loading control. Numbers below the GRP78 bands represent relative levels of GRP78 and are derived from the ratio of GRP78 to β-actin.

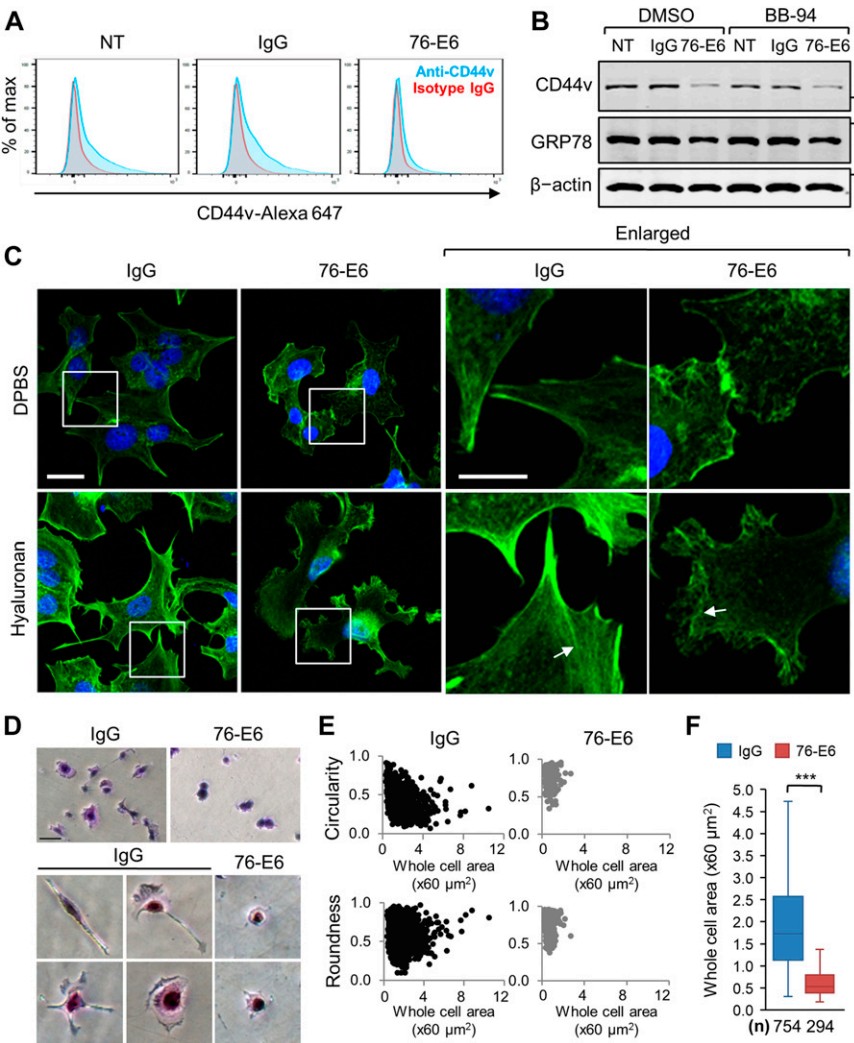

**Figure 5. Antibody against GRP78 (76-E6) reduces CD44v protein level and suppresses cell spreading in MCF7-LR cells.**

**(A)** Flow cytometry histograms representing the levels of CD44v in the NT MCF7-LR cells or cells treated with the rat anti-GRP78 antibody (76-E6) or control IgG for 48 h. Culture plates were precoated with collagen I (100 $\mu$g/ml). CD44v was detected by the anti-CD44v3 antibody. % of max: percentage of maximum staining intensity. Cyan, anti-CD44v; red, isotype control IgG. Cell number analyzed in each group was about 10,000. **(B)** Western blot analysis of whole-cell lysates prepared from NT MCF7-LR cells or cells treated with the 76-E6 antibody or control IgG for 24 h. The broad-spectrum metalloproteinase inhibitor (BB-94) or vehicle control (DMSO) was added into the culture medium together with the antibody groups. CD44v was detected by the anti-CD44v3 antibody. GRP78 was detected by the MAb159 antibody. $\beta$-actin served as a loading control. **(C)** Immunofluorescence and confocal micrographs showing the morphology and F-actin organization of MCF7-LR cells treated with 76-E6 or control IgG for 24 h followed by treatment with hyaluronan or Dulbecco's phosphate-buffered saline vehicle control for additional 4 h. The arrows indicate F-actin. Boxed regions were enlarged in the right panels. Green, F-actin labeled with ActinGreen 488; blue, nuclei stained by DAPI. Scale bar, 20 $\mu$m. **(D)** Cell spreading assay. Bright-field micrographs showing crystal violet–stained MCF7-LR cells. Cells were treated with 76-E6 or control IgG for 24 h and then re-seeded onto collagen I–coated culture plates for 7 h. Representative cells were enlarged in lower panels with same magnification. Scale bar, 20 $\mu$m. **(E)** Circularity, roundness, and area of cells from panel (D) were quantified by the FIJI-Image J software. Circularity = $4\pi \times$ Area/Perimeter$^2$. Roundness = $4 \times$ Area/($\pi \times$ Major axis$^2$). Cell number (n) analyzed in the study: n = 754, IgG; n = 294, 76-E6. **(F)** Individual cell area from panel (D) was calculated by the FIJI-Image J software and represented by the box and whisker plot. n, number of analyzed cells. ***$P < 0.001$ ($t$ test).

added the broad-spectrum MMP inhibitor, BB-94, into culture media together with the 76-E6 treatment and found the level of CD44v reduction was similar to the DMSO control (Fig 5B). Therefore, the observed reduction of CD44v was not because of the proteolytic activity of MMPs. To further investigate the underlying mechanism of the reduction of CD44v, we treated MCF7-LR cells with 76-E6 or control IgG and then co-immunostained cells with antibodies against CD44v and Rab5, an early endosome marker. The co-localization of CD44v and Rab5 was visualized using confocal microscopy (Fig S4B). The extent of their co-localization was analyzed by Coloc 2 plug-in (Mander's O.C.) in the FIJI-Image J software, which revealed a modest but statistically significant increase of CD44v endocytosis after 76-E6 treatment compared with the IgG control (Fig S4C). We next investigated if the reduction of CD44v protein level was because of decrease of transcription using RT quantitative PCR (RT-qPCR) and primers targeting common regions or specific variable exons (Fig S4D). Interestingly, 76-E6 treatment resulted in compensatory increase of overall CD44 transcripts, and this is because of elevation of CD44 containing variable exons 3 or 6 but not CD44 standard (CD44s) isoform (Fig S4E).

Functional analyses showed that 76-E6 treatment caused disorganized F-actin and suppressed hyaluronan, a CD44 ligand, induced F-actin formation (Fig 5C, arrows) compared with the IgG control. The 76-E6 treatment also led to about 60% reduction in cell attachment, as calculated by the ratio of the cells left on the culture plate in the cell spreading assay (Fig 5D; 76-E6/IgG = 294/754). Single-cell morphological analyses using the FIJI-Image J software further showed that 76-E6 treatment resulted in less cell protrusions (higher circularity score), rounder cell shape (higher roundness score), and reduced capability of cell spreading (Fig 5D–F). Of note, 76-E6 treatment did not led to significant cell death within the time frame of our experiments compared with the IgG control (Fig S4F).

Although the 76-E6 antibody (Abcam) was marketed as anti-GRP78 antibody, we surprisingly discovered that this antibody could recognize both recombinant GRP78 (HSPA5) and HSP70 (HSPA1A) in Western blot analysis (Fig S5). It has been reported that cell surface HSP70 in cancer cells largely exists as an integral protein and only a minimal sequence (aa 450–461) close to the COOH-terminal substrate binding domain is exposed outside plasma membrane and accessible by the antibodies (Multhoff & Hightower, 2011). Results

from epitope mapping of the 76-E6 antibody on recombinant GRP78 protein and sequence comparison with the HSP70 protein revealed that the epitope of the 76-E6 antibody on HSP70 was localized outside the region reported to be exposed on the outer leaflet of plasma membrane of the integral HSP70 (data not shown). These data suggest that 76-E6 likely majorly targets GRP78 on the cell surface because in contrast to HSP70, csGRP78 largely exists as a peripheral protein localized outside the cell surface (Tsai et al, 2015).

To further confirm and investigate csGRP78-specific functions, we used a mouse monoclonal anti-GRP78 antibody (MAb159) identified through hybridoma screening to treat MCF7-LR cells. Previously, we have shown that MAb159 specifically recognizes GRP78 but not HSP70 and is capable of blocking lung and liver metastasis in a 4T1 orthotopic breast cancer model (Liu et al, 2013). Similar to treatment with 76-E6, MAb159 treatment resulted in substantial reduction of CD44v compared with the IgG control as demonstrated by IF staining and confocal microscopy in nonpermeabilized MCF7-LR cells (Fig 6A).

The CD44v signal from compressed Z-stack confocal images covering whole cells was quantified by the FIJI-Image J software, and the cells treated with MAb159 (n = 209) exhibited about 50% reduction of CD44v protein level compared with the IgG control (n = 377) (Fig 6A). Furthermore, the MCF7-LR cells treated with MAb159 showed reduced ability to attach to collagen I–coated culture plate (Fig 6B) and decreased capability to spread (Fig 6C and D). These results confirmed the observed phenotypes of MCF7-LR cells treated with siRNAs or 76-E6 and suggest that csGRP78 is a regulator of CD44v membrane homeostasis, cell adhesion, and cell spreading.

## Discussion

Although both GRP78 and CD44 have been widely implicated in aggressive cancer growth and therapeutic resistance, the physical and functional interactions of these two proteins in the context of

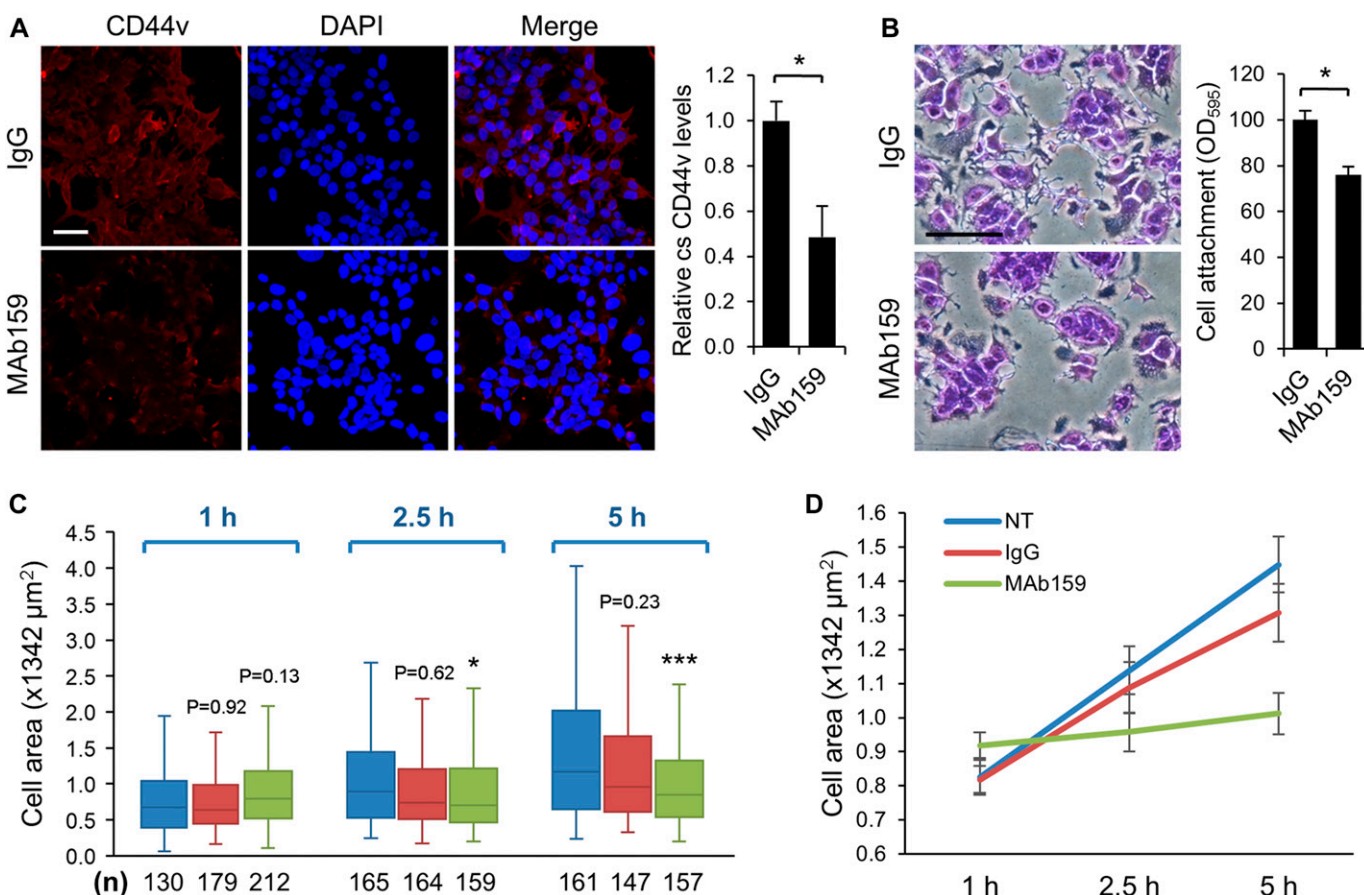

**Figure 6. Antibody against GRP78 (MAb159) reduces CD44v protein level, cell attachment, and cell spreading in MCF7-LR cells.**
**(A)** Left: Immunofluorescence and compressed z-stack confocal micrographs showing the CD44v (red) levels on the cell surface of nonpermeabilized MCF7-LR cells treated with the MAb159 antibody or control IgG for 72 h. The cells were seeded on coverslips coated with collagen I (100 $\mu$g/ml). CD44v was detected by the anti-CD44v3 antibody. Nuclei were stained by DAPI in blue. Scale bar, 20 $\mu$m. Right: Quantification of relative cell surface (cs) CD44v levels. Number of analyzed independent image areas (A) and cells (N): A/N = 5/377, IgG; 3/318, MAb159. Data represent mean ± SEM *P < 0.05 (t test). **(B)** Left: Bright-field micrographs showing crystal violet–stained MCF7-LR cells. Cells were treated with MAb159 or control IgG for 72 h and then re-seeded onto culture plates coated with collagen I (100 $\mu$g/ml) for 1 h. Scale bar, 20 $\mu$m. Right: Quantification of relative levels of cell attachment. Crystal violet staining of adherent cells was dissolved in 100% methanol, and the OD was measured at 595 nm. Data represent mean ± SEM from three biological repeats. *P < 0.05 (t test). **(C)** Kinetic measurement of cell spreading area. MCF7-LR cells were treated with MAb159 or control IgG for 72 h before re-seeding onto collagen I–coated culture plates for the indicated times. Cells were stained with rhodamine phalloidin and visualized by epifluorescent microscopy. Cell area was quantified by the FIJI-Image J software, and the results were represented by the box and whisker plot. *P < 0.05, ***P < 0.001 (t test). **(D)** Data obtained from panel (C) were represented by mean ± SEM for each condition. n, number of analyzed cells.

breast cancer cells resistant to hormonal treatment are just emerging (Chiu et al, 2013; Tseng et al, 2019). Because acquired tamoxifen resistance of breast cancer cells is accompanied with an elevated level of csGRP78, the tamoxifen-resistant MCF7-LR breast cancer cells represent a clinically relevant model to study the interaction of csGRP78 with its partner proteins. Here, we provide evidence that csGRP78 co-expressed and co-localized with CD44v in MCF7-LR cells and in CTCs derived from breast cancer patients. We further established that GRP78 can directly bind to and interact with the extracellular domain of the variant isoform of CD44. Our studies uncovered previously unidentified functions of csGRP78 in regulation of CD44 homeostasis, cell adhesion, and cell spreading, and our findings could have important therapeutic implications for blocking CD44v functions by targeting its partner protein in tamoxifen-resistant breast cancer.

It is well established that GRP78 is a key chaperone in the ER, facilitating folding and maturation of proteins destined for the plasma membrane or to be secreted. We recently observed co-localization of GRP78 and CD44v both in the ER compartment and on the cell surface (Tseng et al, 2019). These results suggest that CD44v could be a client protein of GRP78 during its synthesis in the ER, which could result in their co-trafficking to the cell surface. The ability to study protein diffusion on a subdiffraction limit scale provides opportunities to highlight interactions between GRP78 and CD44v on the cell surface. Using dual-color sptPALM, we discovered the correlated lateral diffusion and co-confinement of GRP78 and CD44v in plasma membrane nanodomains, indicating that they dynamically interact with each other in tamoxifen-resistant breast cancer cells. Our studies revealed that GRP78 knockdown differentially impacts the diffusive behaviors of fast and slow subpopulations of CD44v. This suggests that GRP78 and CD44v could form diverse complexes in plasma membrane nanodomains. Previous studies have shown that CD44 can localize to lipid raft or nonraft membrane microdomains. For example, the association of CD44 with lipid rafts activates SRC family protein kinase and annexin II signaling, thereby regulating cytoskeletal dynamics (Ilangumaran et al, 1998; Oliferenko et al, 1999; Lee et al, 2008). Induction of cell migration in triple-negative MDA-MB-231 breast cancer cells led to reduced affiliation of CD44 with lipid raft, and this was accompanied by increased association of CD44 with active ezrin, a membrane-cytoskeleton linker, in the nonraft fraction (Donatello et al, 2012). The VEGF-induced cell migration of mesenchymal stem cells rapidly modified the nanodomain size of CD44, which resulted in FAK activation and rearrangement of cytoskeleton (Ke et al, 2015). It is tempting to speculate that GRP78 facilitates differential CD44v signaling and actin remodeling by associating with subpopulations of CD44v on the cell surface, and future studies will be required to elucidate these intriguing observations.

Our discovery that targeting csGRP78 resulted in reduction of CD44v protein level on the cell surface raises the question concerning the potential mechanisms. Here, we determined that CD44 reduction is not because of decrease of *CD44v* mRNA levels, rather we observed an increase. This increase in mRNA level is likely a compensatory feedback response to the reduction of CD44v protein level. Another possibility for decrease in the cell surface expression of CD44v could be an increase in the endocytosis of CD44v, which

was observed in MCF7-LR cells treated with an anti-GRP78 antibody. Other potential mechanisms include alteration of CD44v protein stability or its shedding from the plasma membrane, although treatment of MCF7-LR cells with BB-94, an inhibitor of broad-spectrum matrix metalloprotease known to cleave CD44 (Chetty et al, 2012), showed little or no effect on rescuing the CD44v level.

Targeting CD44 using peptides or antibodies has drawn great attention in cancer therapeutics but challenges remain because of abundant expression of CD44 in normal tissues including bone marrow, liver, spleen, and skin (Jin et al, 2006; Marangoni et al, 2009; Masuko et al, 2012; Li et al, 2014; Jordan et al, 2015). Therefore, it is critical to identify alternative approaches to target CD44, possibly through indirect means. Here, we discovered that the molecular chaperone GRP78 interacts with the transmembrane protein CD44v, which could serve to anchor GRP78 on the cell surface of breast cancer cells. Importantly, GRP78 is up-regulated and preferentially expressed on the cell surface of tumor cells and minimally in normal cells, making it an attractive target for cancer-specific therapy, including aggressive breast cancer (Arap et al, 2004; Sato et al, 2010; Liu et al, 2013; Lee, 2014; Dobroff et al, 2016; D'Angelo et al, 2018). Here, we discovered that the antibodies against csGRP78 resulted in reduction of CD44v protein level and suppression of cell spreading capability. These results suggest that perturbation of csGRP78 could represent a previously unidentified strategy for anti-CD44 therapy, which warrants vigorous future investigation.

## Materials and Methods

### Cell culture and reagents

Tamoxifen-resistant MCF7 (MCF7-LR) cells were gifts from Dr Rachel Schiff (Baylor College of Medicine) and cultured in phenol red–free RPMI containing 5% charcoal-stripped fetal bovine serum, 200 mM glutamine, 2.5 µg/ml fungizone, 10 IU/ml penicillin, 10 µg/ml streptomycin, and 100 nM 4-hydroxy tamoxifen. HEK 293T and MDA-MB-231 cells were cultured in DMEM containing 10% fetal bovine serum, 4 mM L-glutamine, 4.5 g/l glucose, 100 IU/ml penicillin, and 100 µg/ml streptomycin. Patient-derived CTCs were cultured as described (Yu et al, 2014). All non-CTCs were authenticated by STR DNA profiling analysis at the Bioreagent and Cell Culture Core Facility in the USC Norris Comprehensive Cancer Center. Only mycoplasma-negative cells were used. Antibodies used in the study are listed in Table S1. Reagents used in the study are listed in Table S2.

### Plasmids and cloning

The construction of plasmids including CD44v-HA, GST, and GST-tagged GRP78 (FL, N, and C) was previously described (Tseng et al, 2019). The CD44v-EC-His plasmid was produced by PCR amplification of the CD44v3-10 coding sequence (nt 1–1,821) from the CD44v-HA plasmid with the reverse primer containing hexahistidine sequence. The PCR product was inserted into a pcDNA3 vector at KpnI and ECoRI sites. The PATagRFP-GRP78 plasmid was produced by PCR amplification of the PATagRFP coding sequence from an

ABP-PATagRFP expression plasmid using the forward primer containing GRP78 ER signal sequence (78ERSS). The caveolin-1 coding sequence in caveolin-1-SNAP expression plasmid was then replaced by the 78ERSS-PATagRFP sequence at ECoRI and SacII sites. This generated an intermediate 78ERSS-PATagRFP-SNAP expression plasmid. Then, the PATagRFP-GRP78 expression plasmid was produced by PCR amplification of GRP78 coding sequencing from FLAG-tagged human GRP78 (wild-type) expression plasmid. The SNAP coding sequence in 78ERSS-PATagRFP-SNAP expression plasmid was then replaced by the GRP78 coding sequence at SacII and KpnI sites. The SacII restriction site was destroyed after cloning. The CD44v-PAGFP expression plasmid was produced by PCR amplification of the CD44v3-10 coding sequence from the CD44v-HA expression plasmid. The caveolin-1 coding sequence in caveolin-1-SNAP expression plasmid was then replaced by the CD44v3-10 coding sequence at ECoRI and SacII sites. This generated an intermediate CD44v3-10-SNAP expression plasmid. Then, the CD44v-PAGFP expression plasmid was produced by PCR amplification of PAGFP coding sequence from PAGFP-CD4 expression plasmid. The SNAP coding sequence in CD44v3-10-SNAP expression plasmid was then replaced by the PAGFP coding sequence at SacII and KpnI sites. All constructs were verified by sequencing. The primers used in the study are listed in Table S3.

## Plasmid transfection

Cells were transfected with BioT transfection reagent (Bioland Scientific) according to the manufacturer's instruction. Media were freshly replaced 5 h posttransfection. Cells were collected 48 h posttransfection for Western blot analysis or purification of recombinant proteins.

## Gene knockdown

For short interfering RNA (siRNA) knockdown, cells were transfected with Lipofectamine RNAiMAX reagent (Thermo Fisher Scientific) containing siRNA (Dharmacon; GE Healthcare) to the final concentration of 60 nM. Oligonucleotides used in the study are listed in Table S3.

## Immunoblot analysis

Cells were lysed in radioimmunoprecipitation buffer (50 mM Tris–HCl, pH 7.5, 150 mM NaCl, 1% NP-40, 0.5% sodium deoxycholate, 0.1% SDS, and a protease and phosphatase inhibitor cocktail). The cell lysates were subjected to 10% SDS–PAGE and transferred onto nitrocellulose membrane (Bio-Rad Laboratories). The membrane was blocked by TBS containing 0.05% Tween-20 and 5% nonfat dry milk at RT for 1 h and then incubated with primary antibody at 4°C overnight. After four washes, the membrane was incubated with secondary HRP-labeled antibody (Thermo Fisher Scientific and Santa Cruz Biotechnology) or fluorescent IRDye-labeled antibody (LI-COR). HRP signal was detected by chemiluminescence (ECL) and quantified with Image Lab (Bio-Rad Laboratories). Fluorescent IRDye signal was detected by Odyssey (LI-COR).

## Immunofluorescence and confocal microscopy

For detection of endogenous GRP78 and CD44 containing v3 exon on the cell surface of MCF7-LR cells, cells were grown for 48 h to subconfluence on sterile coverslips. Coverslips were coated with 50 µg/ml poly-L-lysine in ultrapure water (Sigma-Aldrich) at RT for 1 h followed by 100 µg/ml collagen I from rat tail (Corning Inc.) in 0.02% acetic acid at RT for 2 h. The cells were fixed in 4% paraformaldehyde (Electron Microscopy Sciences) in Dulbecco's phosphate-buffered saline at RT for 10 min and blocked with 4% BSA in PBS at RT for 1 h. The primary antibody against GRP78 (MAb159) was incubated with the cells at 4°C overnight in blocking buffer, followed by staining with Alexa Fluor 594 or Alexa Fluor 568 secondary antibody (Thermo Fisher Scientific) at RT for 1 h. Then, the cells were treated with M.O.M. mouse Ig blocking reagent (Vector Laboratories) at RT for 2 h to block mouse immunoglobulin from primary mouse anti-GRP78 antibody. The cells were then incubated with the primary antibody against CD44 variable exon 3 (Thermo Fisher Scientific) at 4°C overnight in blocking buffer, followed by staining with Alexa Fluor 488 or Alexa Fluor 647 secondary antibody (Thermo Fisher Scientific) at RT for 1 h. Each step was followed by four washes in PBS. Coverslips were mounted with Vectashield antifade medium containing DAPI (Vector Laboratories), and the fluorescent signals were visualized on a Zeiss LSM510 confocal microscope (Carl Zeiss). Z-stack images were acquired with a Plan-Apochromat 100×/1.4 NA oil DIC objective.

For detection of endogenous GRP78 and CD44 containing v3 exon on the cell surface of BRx-68 and BRx-07 CTCs derived from breast cancer patients, cells in suspension were immunostained using the protocol described above. Before mounting with Vectashield antifade medium containing DAPI (Vector Laboratories), cells were rinsed with ultrapure water (Sigma-Aldrich) and applied to Superfrost Plus micro slide (VWR International). The fluorescent signals were visualized on a Zeiss LSM510 confocal microscope (Carl Zeiss). Z-stack images were acquired with a 63×/1.4 NA oil immersion objective.

## Single-particle tracking and diffusion analyses

For dual-color single-particle tracking by PALM, MCF7-LR cells were seeded on Marienfeld-Superior precision coverslips (thickness no. 1.5H) coated with collagen I (100 µg/ml) and transfected with PATagRFP-GRP78 and CD44v-PAGFP expression plasmids 24 h after seeding. For sptPALM of CD44v-PAGFP with GRP78 knockdown, MCF7-LR cells seeded on collagen I–coated Marienfeld coverslips were co-transfected with the CD44v-PAGFP expression plasmid and siRNA using BioT transfection reagent (Bioland Scientific) for 5 h. The culture media was replaced by media containing Lipofectamine MessengerMAX reagent (Thermo Fisher Scientific) and siRNA oligos. 48 h after initial transfection, imaging was performed by TIRF on an inverted Nikon Eclipse Ti-E microscope, equipped with a 100×/1.49 NA objective (Nikon), two iXon EMCCD cameras (Andor Technology), a dual camera light path splitter (Andor Technology), an axial stabilizing system (Perfect Focus System; Nikon), and laser lines at 405, 488, and 561 nm (Agilent). A multiband pass ZET405/488/561/647x excitation filter (Chroma Technology), a quad-band ZT405/488/561/647 dichroic mirror (Chroma Technology), and an emission FF560-FDi01 dichroic mirror (Semrock), and appropriate emission

filters for simultaneous sptPALM imaging of PATagRFP-GRP78 (600/50 nm; Chroma Technology) and CD44v-PAGFP (525/50 nm; Chroma Technology) were used. In both emission channels, images were acquired continuously at a frame rate of 30 ms/frame. Precise image alignment of both RFP and GFP channels was performed using 40 nm TransFluoSphere fiducials (488/685 nm; Life Technologies) spread around the cells.

Single-particle localization and tracking were performed using SlimFast, a single-molecule detection and tracking software written in Matlab. Localizations were done by 2D Gaussian fitting of the point-spread function of each activated PATagRFP-GRP78 and CD44v-PAGFP in each frame. Diffusion trajectories were built by linking individual localized positions from one frame to the other, taking into account blinking statistics and local particle densities. Only trajectories with at least three steps were kept for diffusion analyses based on the MSD of individual trajectories or on the PDSD (Fernandez et al, 2017) of all trajectories. Diffusion coefficients from individual MSD were determined by fitting MSD curves over the first three time lags using a free Brownian diffusion model with measurement error:

$$r^2 = 4Dt + 4\sigma^2, \tag{1}$$

where $\sigma$ is the position error and $D$ is the diffusion coefficient.

Diffusion coefficients from PDSD were determined by fitting each $Pr^2$ curve over the first 10 time lags with the general model:

$$P\left(\overrightarrow{r}^2, t\right) = 1 - \sum_{i=1}^{n} \alpha_i(t) e^{-r^2/r_i^2(t)} \tag{2}$$

$$\sum_{i=1}^{n} \alpha_i(t) = 1,$$

where $r_i^2(t)$ and $\alpha_i(t)$ are the square displacement and the fraction corresponding to $i$ numbers of diffusive behaviors at each time lag $t$, respectively. For both PATagRFP-GRP78 and CD44v-PAGFP, the $Pr^2$ distributions were best fit with $i = 2$ behaviors. Error bars for each $r_i^2$ in $r_i^2(t)$ curves were determined using $\frac{r_i^2}{\sqrt{N}}$, where $N$ is the number of data points used to build each probability distribution function. Diffusion coefficients were obtained by fitting $r_i^2(t)$ curves with an Origin software (OriginLab) and using the free Brownian diffusion model with localization error in Equation (1) or using a circularly confined diffusion model with measurement error:

$$r^2 = R^2\left(1 - A_1 e^{-\frac{4A_2Dt}{R^2}}\right) + 4\sigma^2, \tag{3}$$

where $R$ is the confinement radius, $\sigma$ is the position error, $D$ is the diffusion coefficient, $A_1 = 0.99$ and $A_2 = 0.85$ (Pinaud et al, 2009).

All the diffusion coefficients $D$ are reported in micrometer squared per second ± SD of the fit value.

### Flow cytometry analysis

Cells were collected using nonenzymatic cell dissociation solution (Sigma-Aldrich) and blocked with blocking buffer (Dulbecco's phosphate-buffered saline, 3% FBS, and 0.1% sodium azide) on ice for 1 h. For co-expression analysis, $1 \times 10^6$ cells were aliquoted and incubated with the 10 µg/ml primary antibody against GRP78 (MAb159) or corresponding isotype control (BioLegend) on ice for

1 h in blocking buffer, followed by staining with Alexa Fluor 488 secondary antibody (1:200; Thermo Fisher Scientific) on ice for 1 h. Then, the cells were treated with M.O.M. mouse Ig blocking reagent (Vector Laboratories) at RT for 1 h to block mouse immunoglobulin from primary mouse anti-GRP78 antibody. The cells were then incubated with the 10 µg/ml primary antibody against CD44 variable exon 3 (Thermo Fisher Scientific) or corresponding isotype control (BioLegend) on ice for 1 h in blocking buffer, followed by staining with Alexa Fluor 647 secondary antibody (1:200; Thermo Fisher Scientific) on ice for 40 min. Each step was followed by three washes in blocking buffer. For single staining in antibody treatment experiments, $1 \times 10^5$ cells were aliquoted and blocked with M.O.M. mouse Ig blocking reagent (Vector Laboratories) and 3% FBS. Then, cells were incubated with 10 µg/ml anti-CD44v3 primary antibody or corresponding isotype control (BioLegend) followed by Alexa Fluor 647 secondary antibody (1:200). Cells were suspended in blocking buffer containing 1 µg/ml DAPI (Sigma-Aldrich) and subjected to flow cytometry. The data were acquired by LSR II (co-expression analysis) or FACSVerse (single staining) flow cytometer (Becton Dickinson) and analyzed with the FlowJo v10 software.

### Purification of GST-tagged recombinant proteins

Purification of GST-tagged recombinant proteins was performed as previously described (Tseng et al, 2019). Briefly, FL GRP78 and truncated mutants were cloned into pGEX-4T-1 vector and transformed into E. coli (BL21). The expression of the GST fusion proteins was induced with 4 mM IPTG at 37°C and 200 rpm for 4 h. Bacterial cells were lysed in TBS containing 50 mM Tris–Cl, pH 7.5, 150 mM NaCl, 1% Triton X-100, 1 mg/ml lysozyme, and protease and phosphatase inhibitor cocktails (Thermo Fisher Scientific). Supernatant was collected and incubated with glutathione-Sepharose 4B beads (GE Healthcare) at 4°C for 12 h. Recombinant GST-tagged protein was eluted with freshly prepared reduced glutathione (10 mM; Sigma-Aldrich) at 4°C for 12 h. Then, the recombinant protein was buffer-exchanged to TBS. For long-term storage, the recombinant protein in TBS containing 15% glycerol was snap-frozen in liquid nitrogen and then stored at −80°C.

### Purification of polyhistidine-tagged recombinant proteins

The CD44v-EC-His plasmid was transfected into mammalian 293T cells. 48 h after transfection, cells were lysed in PBS buffer containing 50 mM sodium phosphate, 150 mM NaCl, 1% NP-40, and protease and phosphatase inhibitor cocktails (Thermo Fisher Scientific). Clarified cell lysate was pooled with concentrated conditional media and then incubated with TALON cobalt resin (Clontech) at 4°C for 12 h with gentle rotation. Then, beads were washed three times with 100× bed volume of PBS buffer supplemented with 40 mM imidazole (Sigma-Aldrich). The beads were then transferred to gravity flow column and washed three times with 40× bed volume of PBS buffer supplemented with 40 mM imidazole. Polyhistidine-tagged protein was eluted with PBS buffer containing 250 mM imidazole at 4°C for 12 h. Then, the recombinant protein was buffer-exchanged to TBS. For long-term storage, the recombinant protein in TBS containing 15% glycerol was snap-frozen in liquid nitrogen and then stored at −80°C.

### In vitro GST pull-down assay

The GST pull-down assay was performed as previously described (Tseng et al, 2019). Briefly, GST tag or GST-tagged GRP78 (FL, N, and C) was coupled to Glutathione Sepharose 4B beads and then incubated with 1 mg 293T WCL containing overexpressed CD44v-HA at 4°C overnight in IP lysis buffer (Thermo Fisher Scientific; 25 mM Tris–HCl, pH 7.4, 150 mM NaCl, 1% NP-40, 1 mM EDTA, and 5% glycerol) followed by six washes with lysis buffer. Bound proteins were eluted with 2× SDS sample buffer.

### In vitro direct binding assay

GST tag (2.5 µg) or GST-tagged FL GRP78 (2.5 µg) was coupled to Glutathione Sepharose 4B beads at 4°C for 4 h. Then, the beads conjugated with GST or GST-tagged GRP78 were incubated with recombinant hexahistidine-tagged extracellular region of CD44v3-10 (1.5 µg) at 4°C overnight in the binding buffer (50 mM Tris–HCl, pH 7.5, 150 mM NaCl). Beads were then washed four times with the binding buffer containing 0.05% Tween-20. Bound proteins were eluted with 2× SDS sample buffer.

### Real-time quantitative RT–PCR

The MCF7-LR cells were treated with the 76-E6 antibody or the control IgG for 24 h, and then RNA was extracted from the treated cells using the TRI reagent (Sigma-Aldrich). The reverse transcription was performed using 1.5 µg extracted total RNA per condition, random primers (New England Biolabs), and the SuperScript II reverse transcriptase (Thermo Fisher Scientific). The cDNA samples were analyzed with the SYBR Green Supermix (Quanta Biosciences) according to the manufacturer's instructions using the Mx3005P thermocycler (Stratagene). The primers used in the study are listed in Table S3.

### Cell attachment assay

MCF7-LR cells were transfected with siRNAs for 60 h or treated with the antibody against GRP78 (MAb159, 50 µg/ml) or the corresponding control IgG for 72 h. Cells were then dissociated with 0.025% trypsin and 0.01% EDTA in PBS (siRNA group) or nonenzymatic cell dissociation solution (antibody group; Sigma-Aldrich) and plated at a density of 60,000 cells per well on 48-well plates precoated with collagen I (100 µg/ml). Cells were allowed to adhere for 1 h. Adherent cells were fixed with 100% methanol at −20°C for 10 min and then visualized by crystal violet staining (0.5% wt/vol crystal violet in 20% ethanol for 10 min). Crystal violet staining of adherent cells was then dissolved in 100% methanol, and the optical density (OD) of the methanol solution was measured at 595 nm.

### Cell spreading assay

MCF7-LR cells were transfected with siRNAs for 60 h or treated with the antibody against GRP78 (MAb159, 50 µg/ml) or the corresponding control IgG for 72 h. Cells were then dissociated with 0.025% trypsin and 0.01% EDTA in PBS and plated at a density of 30,000 cells per well on 6-well plates or 6,000 cells per well on 12-well plates precoated with collagen I (100 µg/ml). Cells were allowed to adhere for 1, 2.5, or 5 h and then fixed with 4% para-formaldehyde at RT for 10 min. Then, cells were permeabilized with 0.1% Triton X-100 at RT for 1 min followed by staining with rhodamine phalloidin according to the manufacturer's protocol (Cytoskeleton, Inc.). Cells were then visualized by the Keyence fluorescent microscope (BM-X710), and the images were acquired with a 20× objective. The cell area was analyzed by the FIJI-Image J software.

For the MCF7-LR cells treated with the antibody against GRP78 (76-E6, 50 µg/ml) or the corresponding control IgG, cells were treated for 24 h and then dissociated with nonenzymatic cell dissociation solution (Sigma-Aldrich). Then, cells were plated at a density of 30,000 cells per well on 96-well plates precoated with collagen I (100 µg/ml) and allowed to adhere and spread for 7 h. Cells were then fixed with 100% methanol at −20°C for 10 min and visualized by crystal violet staining (0.5% wt/vol crystal violet in 20% ethanol for 10 min). Cell morphology and area were analyzed by the FIJI-Image J software.

### Statistics and reproducibility

Statistical analysis was performed using Microsoft Excel, FIJI-Image J and OriginPro software. Data are presented as mean ± SEM. Significance was calculated by two-tailed unpaired $t$ test or Kolmogorov–Smirnov test. Statistical significance was represented as $*P < 0.05$, $**P < 0.01$, and $***P < 0.001$. All experiments were independently performed three times or with specified sample/repeat numbers described in the figure legends.

# Supplementary Information

# Acknowledgements

We thank Dr Rachel Schiff (Baylor College of Medicine) for the gift of MCF7-LR breast cancer cells, Dat Ha and Daisy Flores Rangel for helpful discussions. We thank the Cell and Tissue Imaging Core of the USC Norris Comprehensive Cancer Center (supported by National Institute of Health [NIH] grant P30 CA014089) and the Cell and Tissue Imaging Core of the USC Research Center for Liver Diseases (supported by NIH grant P30 DK048522) for assistance with confocal microscopy. This study was supported by NIH grant number R01 CA027607 to AS Lee.

### Author Contributions

C-C Tseng: conceptualization, resources, data curation, software, formal analysis, validation, investigation, visualization, methodology, and writing—original draft, review, and editing.
R Stanciauskas: software, formal analysis, investigation, visualization, and methodology.
P Zhang: formal analysis and investigation.
D Woo: investigation.

K Wu: investigation.

K Kelly: resources.

PS Gill: resources.

M Yu: resources.

F Pinaud: resources, data curation, software, formal analysis, validation, investigation, visualization, methodology, and writing—original draft, review, and editing.

AS Lee: conceptualization, resources, data curation, supervision, funding acquisition, methodology, project administration, and writing—original draft, review, and editing.

## Conflict of Interest Statement

The authors declare that they have no conflict of interest.

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
