## [Reviewer comments · Life Science Alliance]

Life Science Alliance

GRP78 regulates CD44v membrane homeostasis and cell spreading in tamoxifen-resistant breast cancer

Amy Lee, Chun-Chih Tseng, Ramunas Stanciasauskas, Pu Zhang, Dennis Woo, Kaijin Wu, Kevin Kelly, Parkash Gill, Min Yu, and Fabien Pinaud

DOI: <https://doi.org/10.26508/lsa.201900377>

Corresponding author(s): Amy Lee, University of Southern California

Review Timeline:

Submission Date:	2019-03-12
Editorial Decision:	2019-03-13
Revision Received:	2019-07-29
Editorial Decision:	2019-08-01
Revision Received:	2019-08-02
Accepted:	2019-08-05

Scientific Editor: Andrea Leibfried

Transaction Report:

Please note that the manuscript was previously reviewed at another journal and the reports were taken into account in the decision-making process at Life Science Alliance.

Reviewer #1 Review

Comments to the Authors (Required):

In this work, the authors identified a novel interaction between the ER chaperon protein GRP78 and the membrane associated receptor CD44v. This relevance of this interaction is shown through co-ip, GST- pull down and dual color single molecule dynamic imaging. Functionally, GRP78 is proposed to interact with CD44v at the cell surface and to exert some of its function through the latter molecules.

There are some interesting novel findings that do seem to be of interest for the wide cell biological audience of this journal. There is, however, the need to improve the manuscript in various aspects. Major limitations of this manuscript are the lack of a proper set of controls and of mechanistic information as to how GRP78 may influence specific pathways downstream of CD44v and their biological consequences. As such the paper would not seem a strong candidate for publication.

Specific points

Figure 1. Cell spreading: the spreading assay is carried out at steady state after cell have completed their spreading. While the difference in the morphology of MCF7-L and MCF7-LR is apparent, it would be informative to analyze the dynamics of cell spreading morphology also after silencing of gRP78 and CD44v.

It is stated that MCF7-L and MCF7-LR have different levels of GRP78 (references to support this tenet are indeed provided). It would be relevant, however, to silence GRP78 both in MCF7-L and LR and compare the effect on cell spreading and migration. One expectation being that in the former cells (expressing low levels of GRP78) the effect of its downregulation might be marginal. The authors further show that directed cell migration (wound healing assays) is impaired by reduction of GRP78. Notably the reduction of GRP78 is partial at best. Is this due to inefficient siRNA or to elevated stability of GRP78 proteins?. Testing the mRNA levels of GRP78 may provide some hints in addressing this issue.

Also, in fig, 1E-F, it is shown that siGRP78 impact on single cell morphology. It is, however, unclear whether this perturbation also impacts on individual random migration as one might expect given the altered F-actin organization.

To state that GRP78 loss causes altered contractility it would be relevant to show the dynamics of these cells or perform staining with pMLC.

Finally, all these and subsequent experiments are done using one single oligo against GRP78. More oligos and/or reconstitution experiments would be mandatory.

Fig. 2. It is unclear why the authors do not bring along in their analysis also the MCF7-L line to be compared to MCF7-LR, in all the assays shown in this and previous figure.

Figure 3A. The authors by analyzing through IF the distribution of csGRP78 and CD44 stated that the two protein display a preferential front-polarized localization in presumably migrating cells. However, the two proteins appear to be excluded from the front edge and lamellipodia. Additionally, and particularly in MDA-MB-231, there is accumulation of csCD44V at the back of the cells rather than the front. Images at higher resolution, quantification of the relative distribution of csGRP78 and CD44v would be needed to support the contention that the two protein are polarized in migrating cells. This analysis should also include independent markers of polarity (e.g one can orient the cells based on the relative position of the nuclei and the golgi)

Fig. 4. The interaction studies are clear and convincing. One issue to clarify is how and where GRP78 can interact with surface exposed CD44v. In other word, is this interaction occurring primarily in the ER along the secretory/maturation path of CD44v or exclusively on the cell surface?

Figure 5 shows that the use of Dual-color single particle tracking to support the possibility of cell surface interaction between GRP78 and CD44V. There are, however, a number of controls that would appear necessary to back up the data in this figure. First, what are the slow and fast diffusion coefficient of a generic fluorescent protein targeted to the PM as compared to GRP78 and CD44v? This appear particularly relevant since the extent of membrane colocalization of the two proteins is limited.

Second, the authors identified the region of GRP78 that interact with CD44v what are the dynamic behaviors of mutants of GRP78 that no longer associate with CD44v?

Finally, the authors analyzed CD44v dynamics after silencing of GRP78. It is expected that the dynamics of GRP78 should also be impacted by silencing of CD44v. Is this the case?

In this as well as in all the experiments where GRP78 siRNA are used it would be necessary to use more than oligo and ideally to reconstitute the expression of GRP78 with WT and mutant variants.

In Figure 6, relevantly, the authors identified an apparently inhibitor antibody against GRP78. To strengthen the specificity and efficacy of this ab, the same antibody should be applied on cells silenced for GRP78. Additionally, the authors should assess whether the phenotypes elicited by 76-E6-addition to those caused by the loss of GRP78 and CD44v. For example, why does the addition of 76-E6 inhibit gelatin degradation? Is this phenotype seen also upon depletion of GRP78 and CD44v? Is this phenotype linked/interference to/with loss of GRP78 or Cd44v?

In Figure 7, the effect of adding 76-E6 is shown on single cell random motility. In Figure 1, however, the authors analyze collective directed motion after removal of GRP78. It would be relevant to run single and directed cell assays after addition of 76-E6 and GRP78 silencing. (again, an experiment where 76-E6 is added to GRP78 silenced cells should be included as necessary control for the specificity of the antibody).

Figure 8. The interaction between IQGAP and GRP78 is interesting, however, more experiments would need to be done to assess whether the interaction is physiologically relevant, and as to how it might impact on the signaling emanating from the GRP78::CD44v. In this set of experiments which are overall much underdeveloped it is also shown that IQGAP is lost from the leading edge after treatment with 76-E6 abs (8B). However, the images are not extremely convincing and the same analysis should be repeated after GRP78 KD? In other word, it is unclear in the absence of further data what is the role if any of IQGAP in the process triggered by GRP78::CD44v interaction.

Fig. 9. In 9A, it is shown that GRP78 may signal through CD44v in part by activation of STAT3. However, the data presented are somewhat preliminary, and rudimentary. Additionally, a number of interpretation can be offered by the apparent reduction of STA3 phosphorylation after 76-E6 treatment. In the end, we are left with a very unclear picture of how GRP78 may influence the signals emanating from CD44v and what are the biological consequences of perturbation of these pathways on specific cellular processes.

Reviewer #2 Review

Comments to the Authors (Required):

This manuscript described a direct interaction between GRP78 and CD44 on the surface of breast cancer cells. The authors proposed that this interaction regulates actin reorganization and dynamics, thus affects cell adhesion, spreading migration, matrix degradation, etc. It was also proposed that GRP78 regulates the homeostasis of CD44 on the plasma membrane by binding to CD44 and suppressing CD44 endocytosis. By this means, GRP78 helps maintain the signaling downstream of CD44. CD44v isoforms are important cell surface markers for cancer stem cells, and likely play critical roles in regulating the cancer stem cell properties including self-renewal, tumor initiation, metastasis and chemoradioresistance. However the underlying molecular mechanisms are unclear. GRP78 as a widely studied ER stress markers, was recently found translocating to the plasma membrane in many types of cancer cells and contributing to cancer survival and treatment resistance. However, the molecules and pathways endorsing these functions to cell surface GRP78 have not been clearly defined. It is very interesting that these two proteins physically interact with each other. However, results presented in current manuscript are rather descriptive although many related cellular events and signaling molecules/pathways were checked, making it hard to interpret the working mechanism.

Main points:

1. If the authors want to state that GRP78 regulates actin re-organization, cell adhesion, migration, and so on by binding to CD44, they need to create GRP78 mutant that cannot bind CD44 and CD44 mutant that cannot bind GRP78, and then use these tools to test their hypothesis.
2. In cancer cells including the two breast cancer cells used in this manuscript, there are many receptors and pathways regulating actin dynamics and migration. How could these molecules, for example integrins, can be excluded from being co-receptors, regulators, or downstream effectors of GRP78 during GRP78-dependent actin organization and cell migration?
3. Does GRP78 function upstream or downstream of CD44? From Fig. 6C seems like GRP78 is downstream of CD44? But from Fig. 9D seems like GRP78 is upstream of CD44?
4. Would 76-E6 binding to GRP78 decrease the binding of CD44 to GRP78?
5. It was shown that GRP78 binds to the ecto-domain of CD44, but also binds to IQGAP1. Is GRP78 a transmembrane protein? Does GRP78 bind to the cytoplasmic domain of CD44?
6. All cell biological microscope imaging data and immunoblotting data showing increase/decrease should be quantified and statistically analyzed from at least three independent experiments, for imaging results >50 cells should be included in each experiment.
7. What is the physiological significance of the slow and fast diffusion of CD44? How would the 76-E6 treatment change the slow and fast diffusion of CD44 and how this may relate to the signaling and cell behavior changes?

Minor points:

1. The molecular weight markers seem like labeled wrong in Fig. 4E, when comparing with the GST-GRP78 fragments shown on other blots like 4C.

Reviewer #3 Review

Comments to the Authors (Required):

In this work, Tseng, et al., report an association between cell-surface associated Grp78 (csGrp78) and the membrane protein CD44v. In their work, they show that a fraction of the Grp78 chaperone pool leaves the ER lumen and remains associated with the plasma membrane in breast cancer cells. The authors also claim that an association between csGrp78 and CD44v occurs on plasma membrane nano-domains in breast cancer cells and that such interaction exerts regulatory control on CD44v signaling, ultimately affecting cancer cell adhesion, spreading and migration. The authors suggest that blocking the Grp78-CD44v interaction negatively impacts cell adhesion, spreading and migration, and therefore targeted intervention aimed at inhibiting csGrp78 may be a viable therapeutic for aggressive breast cancer. While the work showcases an exciting and interesting finding, namely the interaction between csGrp78 and CD44 and its potential roles in cancer biology, the manuscript unfortunately falls short in several aspects, which are detailed below. For this reason, at this moment I cannot recommend publication of this work in this journal, unless it is meticulously revised and potential mechanisms are addressed.

Major points:

1. The authors identify a major ~250 kDa CD44v variant in breast cancer cells. However, when they tag the protein and express in different cells (HEK293T) they produce a ~130 kDa protein. They attribute the large difference in molecular weight to post-translational modifications found in

different cells. While this explanation may be satisfactory, the authors should address such large difference in molecular weight more specifically. The authors write in page 11 "CD44 is one of the most heavily post-translationally modified protein at the plasma membrane", is it not possible that some of the physiological effects they observed could be directly related to such modifications, for instance, in controlling the association of CD44v with csGrp78? Such a possibility falls within the scope of the work and should be appropriately addressed.

2. In figure 5, the authors express fluorescently tagged Grp78 and CD44v use them in TIRF/PALM experiments aimed at investigating co-localization. According to their construct diagrams, the fluorescent proteins would reside in opposite sides of the plasma membrane (for csGrp78, TagRFP would reside in the ER lumen or on the cell surface, whereas for CD44v, the C-terminal PA-GFP tag would be located in the cytosol). Therefore, it is hard to reconcile co-localization on the plane of the membrane or co-confinement in membrane nano-domains from this experiment. The observed interactions could also represent ER-plasma membrane contact sites as TagRFP-Grp78 is not confined to the cell surface or to the ER lumen. Release of TagRFP-Grp78 from the ER lumen would result in its secretion or plasma membrane association as a peripheral membrane protein (as pointed out by the authors), and thus the calculated diffusion rates of TagRFP-Grp78 would not reflect those of TagRFP-Grp78 alone, but those of a yet-to-be-defined complex containing scGrp78. More conclusive results could have been obtained from a FRET-based experiment instead.

3. In figures 6-9, the authors base most of their claims on the results obtained from using a single anti-Grp78 antibody, clone 76-E6. While treatment of live cells with such an antibody results in suppression of phenotypes associated with cancer progression (e.g., adhesion, spreading, motility), more experiments should have been done to address specificity. For instance, using RNAi the authors show they have genetic control over Grp78 levels; one would expect the results from the antibody-based experiments to phenocopy those obtained in RNAi experiments. While it is encouraging that the experiments with SubA seem to support a role for Grp78 in the phenotypic changes reported, one would also expect other antibodies against Grp78 to show similar results. Such experiments would greatly strengthen the conclusions.

Minor points:

1. Please show siRNA/shRNA specificity for Grp78 and lack of off-target effects towards other Hsp70-type proteins by western blot/RT-PCR.
2. Is the UPR dysregulated in the breast cancer cells upon Grp78 knock-down? If so, how would UPR signaling impinge on cellular morphology, adhesion, migration?
3. Is CD44 a folding substrate of Grp78? If so, one would expect that decreasing Grp78 levels would result in decreased CD44 levels, which in turn could explain the observed phenotypes. While the experiments targeting csGrp78 with a neutralizing antibody substantiate the role of csGrp78 in CD44 signaling, one cannot discard the former possibility when conducting gene depletion experiments.
4. Referring to figure 2A, the authors mention that the CD44 isoforms identified by RT-PCR were sequenced. Please provide the data supporting the identification of specific isoforms.
5. In figure 3A, the authors describe the co-localization of CD44v and csGrp78 as "prominent"; and while accurate for MCF7-LR cells, the micrographs for MDA-MB-231 cells do not show prominent co-localization of both proteins. Similarly, the authors claim that the proteins preferentially co-localize to the migrating leading edge of the cells, which is not apparent in the micrographs for

MCF7-LR cells. The statement is subjective as it is based on the qualitative assessment of the micrographs alone and no metrics for such claim are presented.

6. In figure 4, it would have been very useful to see a blot with anti-HA antibody to be able to distinguish endogenous from ectopically expressed CD44 variants; or is it that HEK293T do not express it in the first place?

7. The subsection titled "CD44 containing exon v3 is a major mediator of CD44 functions in aggressive breast cancer" is developed exclusively from supplementary data and not from main findings. Please consider merging with another section or moving supplementary data to main figures to substantiate the point.

8. In figure 5C, the authors show expression of tagged CD44v in HEK293T cells. However, all their TIRF/PALM experiments were conducted in MCF7 breast cancer cells. Please show that ectopic expression of tagged CD44v in MCF7 cells produces the same protein. Or is it heavily post-translationally modified in MCF7 cells?

9. In figure 8, the authors identify IQGAP1 as a major physical interactor of csGrp78. Experiments in which the authors blocked csGrp78 with the 76-E6 antibody were conducted to address the relevance of such interaction and the role of CDC42. However, no additional experiments aimed at linking signaling to CD44v were performed, leaving the reader with a sense of lack of connectivity to the main story line in the manuscript. What is the connection to CD44 signaling?

10. In figures 9A-C the authors claim that the observed reduction in CD44 levels is not attributable to extracellular proteolysis, endosomal recycling or transcriptional regulation. What it is due to? Such mechanism is not addressed in the manuscript. An alternative explanation is that CD44 protein levels changed due to differences in the rate of protein synthesis. Simple pulse-chase experiments could have been done to address one or the other possibility.

11. In figure 9D the authors state that treatment with the 76-E6 antibody results in suppression of STAT3 signaling. However, their data also show a reduction in STAT3 protein levels (lane 3).

12. Please provide a consistent nomenclature for the "CD44v" isoform under study throughout the text.

13. In page 5, please explain what do you refer to as the "CD44 standard isoform".

14. For the micrographs in figure 1, please indicate in the text whether to follow arrows or arrowheads as done in other figures.

15. On page 7, please consider changing "Identification of CD44v as a novel partner protein of csGrp78" to "CD44 and csGrp78 co-localize" as the data shown in figure 3 corresponding with this section do not show physical interaction.

March 13, 2019

Re: Life Science Alliance manuscript #LSA-2019-00377-T

Dr. Amy S. Lee
University of Southern California
USC Norris Comprehensive Cancer Center
1441 Eastlake Avenue, NOR 5307
Los Angeles, CA 90033

Dear Dr. Lee,

Thank you for transferring your manuscript entitled "GRP78 binds CD44v and regulates its membrane homeostasis and signaling in aggressive breast cancer" to Life Science Alliance. The manuscript was assessed by expert reviewers at another journal before and the editors transferred those reports to us with your permission.

The reviewers thought that your work is interesting, but noted some missing controls, quantifications and statistics and they thought that the mechanistic insight offered remains limited. The latter is not a concern for publication in Life Science Alliance, and I would thus like to invite you to submit a revised version of your work to us. The reviewer concerns relating to robustness of the data / lack of support for the conclusions should get addressed:

Rev#1: add control by silencing GRP78 in MCF7-L and MCFLR (pt 1); add assays using another siRNA for GRP78 (point 1); address points relating to figure 3+4 in the ms text; add clarification regarding this reviewer's opinion about figure 5; add control requested in regard to figure 6; perform exp as proposed in concern relating to fig7 to better link the results to figure 1; remove IQGAP/STAT3 data (concerns relating to figs 8+9).

Rev#2: add quantifications and statistics; address minor point 1

Rev#3: add control as requested in point 3; address minor points 1, 4-6, 8 and discuss alternative hypothesis in manuscript text (point 10).

Thank you for this interesting contribution to Life Science Alliance. We are looking forward to receiving your revised manuscript.

Sincerely,

B. MANUSCRIPT ORGANIZATION AND FORMATTING:

Reviewer 1

#1 Add controls to si78 experiments by silencing GRP78 in both MCF7-L and MCF7-LR cells and add another si78 sequence.

Response: As shown in the new Fig. 4A, we now used two siRNAs against GRP78, one against the coding sequence and the other one against the 3'UTR, to confirm our results. Both siRNAs efficiently knockdown the expression of GRP78 by about 70% and have no effect on the expression of HSP70, a related chaperone protein which shares 62% sequence identity with GRP78. The sequence specificity of the siRNAs for GRP78 are shown in the new Fig. 4B. We treated both MCF7-L and MCF-LR cells with these si78 sequences and observed similar morphological changes. Since GRP78 is an essential chaperone protein for all cells, it is not surprising that both cell lines were affected. Therefore, rather than comparing the parental and resistant cells, we focus on using the tamoxifen-resistant MCF7-LR cells which expressed a higher level of GRP78 than the parental MCF7-L cells and co-expressed robust level of CD44 as a model system to study their interactions and functional significance.

With regard to the reviewers' request on detection of phosphorylated myosin light chain levels in these cells, we have tested two anti-phospho-myosin light chain antibodies from Cell Signaling Technology (1) Phospho-Myosin Light Chain 2 (Thr18/Ser19) Antibody #3674 and (2) Phospho-Myosin Light Chain 2 (Ser19) Antibody #3671, both of which have been cited in literature. However, we only detected very low signals in our cells with both antibodies. Therefore, we were unable to quantitate them for inclusion in our results.

#3 Address in MS text regarding GRP78 and CD44v colocalization

Response: We have removed the claim on preferential co-localization at the front region of unipolar cells in the revised text.

#4 Address in MS text regarding where GRP78 can interact with CD44v

Response: We have added discussion of the possible interaction of CD44v with GRP78 in the endoplasmic reticulum as part of the protein folding function of GRP78 in the revised text (p.17).

#5 Address single particle tracking

Response: Due to the complexity of experiments requested and limited resources, we will not be able to perform new experiments using this technique; however, we are able to address in the text the concerns raised.

a) With regard to the need for control with a generic fluorescent protein targeted to the PM, we cited two new publications (see below) that fully confirm our tracking results on CD44. First, we have previously measured the lateral diffusion coefficients for CD4, which like CD44, displayed the same ability to associate with glycosphingolipid-rich plasma membrane (PM) domains at the PM (**Pinaud F. and Dahan M., PNAS, 2011**). As expected, our values determined for CD44 closely resembled that published for CD4. In another recently published study (**Freeman S.A. et al., Cell, 2018**), their results fully confirmed our tracking results on CD44, including the heterogeneous diffusive behaviors of CD44, its slow and fast diffusion coefficients, and its confinement in nanodomains having the same areas that we report. On the contrary, the observed diffusion coefficient of GRP78 is very slow compared to what is expected for a generic FP fusion, regardless of its mode of anchoring at the PM. This suggests that GRP78, which is normally not expressed at the PM, is associated with the membrane in a peculiar fashion (may be clusters), that does not resemble traditional membrane-anchored proteins.

b) With regard to the concern of limited membrane co-localization of CD44 and GRP78, we would like to clarify that it is totally expected that apparent co-localization will be highly infrequent during dual-color sptPALM, simply because the probability of imaging the interaction between 1 out of the thousands CD44v-PAGFP being expressed with 1 out of the thousands PATagRFP-GRP78 being expressed is extremely small. If one considers the fact that endogenous CD44 and GRP78 are also expressed, this low probability is even lower. That does

not mean interactions are infrequent. Indeed, any photoconverted CD44v-PAGFP that we track might be interacting with a non-photoconverted PATagRFP-GRP78 or an endogenous GRP78 and vice versa. These interactions, however will not be observed. The fact that, despite an extremely low probability of detecting interactions, we are still able to detect a few events where CD44v-PAGFP and PATagRFP-GRP78 display coordinated diffusion is a much stronger evidence of interaction than signal co-localization.

c) With regard to the dynamic behaviors of mutants of GRP78 that no longer associate with CD44v, the dynamics of GRP78 in cells after silencing CD44v, and reconstitution of expression of GRP78 with WT and mutant variants in cells treated with siGRP78, *these are interesting experiments to perform but they are huge undertakings and are beyond the scope of the current manuscript.*

#6 Add control to this point (i.e. to strengthen the specificity and efficacy of this ab (**76-E6**), the same antibody should be applied on cells silenced for GRP78.

Response: Considering silencing GRP78 (by si78) alone already elicited severe changes in cell morphology and spreading (revised Fig 4), we suspect it will be very difficult to discern the phenotype differences between the single versus combination treatment with 76-E6. Furthermore, we recently found out the 76-E6 antibody has been discontinued from Abcam, so this experiment is not possible.

#7 Perform experiments mentioned in this point to better link the results to Figure 1.

*Response: We have now utilized another anti-GRP78 antibody (MAb159) to confirm the results that we obtained for the siRNAs against GRP78 (original Fig. 1, now revised Fig. 4) in addition to 76-E6. The MAb159 is a mouse monoclonal antibody which has been validated to specifically recognize GRP78 with no reactivity towards the related chaperone HSP70 (Liu R et al., **Clin Cancer Res 2013**). The new results on MAb159 are presented in the new Fig. 6. In brief, treatment of MCF7-LR cells with either si78, 76-E6 or MAb159 all led to decrease in cell attachment and spreading, providing evidence that csGRP78 is required for the integrity of these cellular functions. Additionally, treatment of cells with either 76-E6 or MAb159 resulted in decrease in cell surface CD44v level.*

#8/#9 Remove IQGAP1/STAT3 data regarding to these points

Response: These data are removed.

Reviewer 2

#6 Add quantifications and statistics

Response: The quantifications and statistics are added (new Fig 4A, right panel; new Fig 4F, G and H; new Fig 5F; new Fig 6A, right panel; Fig 6B, right panel; new Fig 6C and D; new Fig S3B).

Minor point #1 on difference in molecular weight markers labeling

Response: Different protein markers were used since the vendor (New England Biolabs) discontinued the old markers and replaced them with new ones.

Reviewer 3

#3 Add controls for Figures 6-9 (RNAi phenocopy Ab, another Ab)

Response: With our focus on MCF7-LR cells and the essential additional controls, the confirmatory results on MDA-MB-231 cells (original Fig. 7) were removed. We have now utilized another anti-GRP78 antibody (MAb159) to confirm the results that we obtained from the siRNAs against GRP78 (original Fig. 1, now revised Fig. 4) and 76-E6 treatment. The MAb159 is a mouse monoclonal antibody which has been validated to specifically recognize GRP78 with no reactivity towards the related chaperone HSP70 (Liu R et al., Clin Cancer Res 2013). The new results on MAb159 are presented in the new Fig. 6. In brief, treatment of MCF7-LR cells with either si78, 76-E6 or MAb159 all led to decrease in cell adhesion and spreading, providing evidence that csGRP78 is required for the integrity of these cellular functions. Additionally, treatment of cells with either 76-E6 or MAb159 led to decrease in cell surface CD44v level.

Address Minor points (1,4-6,8)

#1 Please show siRNA/shRNA specificity for Grp78 and lack of off-target effects towards other Hsp70-type proteins by western blot/RT-PCR.

Response: As shown in the new Fig. 4A, we now used two siRNAs against GRP78, one against the coding sequence and the other one against the 3'UTR, to confirm our results. Both siRNAs efficiently knockdown the expression of GRP78 by about 70% and have no effect on the expression of a related chaperone protein HSP70 as analyzed by Western blot. The sequence specificity of the siRNAs for GRP78 are shown in the new Fig. 4B.

#4 Referring to figure 2A, the authors mention that the CD44 isoforms identified by RT-PCR were sequenced. Please provide the data supporting the identification of specific isoforms.

Response: The Sanger sequencing results for the identified CD44 isoforms is now included in the supplemental data.

#5 In figure 3A, co-localization of CD44v and csGrp78 overstatement

Response: We have removed the claim on preferential co-localization at the front region of unipolar cells.

#6 In figure 4, endogenous versus ectopically expressed CD44 variants

Response: We have now added a new reference to clarify that 293T cells used to express the CD44 variants was reported to express very low endogenous level of CD44 (Ishimoto, T et al., Cancer Cell, 2011).

#8 In figure 5C, show ectopic expression of tagged CD44v in MCF7 cells

Response: Ectopic expression of HA-tagged CD44v is now shown in the new Fig. 1D.

#10 Discuss alternative hypothesis for CD44 reduction in manuscript text

Response: We have added new discussion on various mechanisms for CD44 reduction such as endocytosis, stability and shedding from the cell surface (page 18-19).

We have addressed all the requested revisions to the best of our abilities, with added controls and quantifications. We hope this revision meets with your final approval of publication in *Life Science Alliance*.

August 1, 2019

RE: Life Science Alliance Manuscript #LSA-2019-00377-TR

Dr. Amy S. Lee
University of Southern California
USC Norris Comprehensive Cancer Center
1441 Eastlake Avenue, NOR 5307
Los Angeles, CA 90033

Dear Dr. Lee,

Thank you for submitting your revised manuscript entitled "GRP78 regulates CD44v membrane homeostasis and cell spreading in tamoxifen-resistant breast cancer". I have now assessed the data introduced in the revision and I think they address the reviewer concerns that we asked you to resolve in a good way. The use of a second antibody against GRP78 is much appreciated, especially in light of the dis-continued 76-E6 one. I think ideally, the effects of the antibody should have been tested in the condition where GRP78 is knocked-down, but I understand your reasoning of not doing so. I would thus be happy to publish your paper in Life Science Alliance. Before sending you the official acceptance letter, please:

- fill in the electronic license to publish form (please also move all files to the next manuscript version when following the link below)
- link your profile in our submission system to your ORCID iD, you should have received an email with instructions on how to do so

A. FINAL FILES:

B. MANUSCRIPT ORGANIZATION AND FORMATTING:

Sincerely,

August 5, 2019

RE: Life Science Alliance Manuscript #LSA-2019-00377-TRR

Dr. Amy S. Lee
University of Southern California
USC Norris Comprehensive Cancer Center
1441 Eastlake Avenue, NOR 5307
Los Angeles, CA 90033

Dear Dr. Lee,

Thank you for submitting your Research Article entitled "GRP78 regulates CD44v membrane homeostasis and cell spreading in tamoxifen-resistant breast cancer". It is a pleasure to let you know that your manuscript is now accepted for publication in Life Science Alliance. Congratulations on this interesting work.

DISTRIBUTION OF MATERIALS:

Again, congratulations on a very nice paper. I hope you found the review process to be constructive and are pleased with how the manuscript was handled editorially. We look forward to future exciting submissions from your lab.

Sincerely,
